# Rapid and precise genome engineering in a naturally short-lived vertebrate

Claire N Bedbrook[1,2†], Ravi D Nath[1†], Rahul Nagvekar[1], Karl Deisseroth[2,3,4], Anne Brunet[1,5,6*]

[1]Department of Genetics, Stanford University, Stanford, United States; [2]Department of Bioengineering, Stanford University, Stanford, United States; [3]Department of Psychiatry and Behavioral Sciences, Stanford University, Stanford, United States; [4]Howard Hughes Medical Institute, Stanford University, Stanford, United States; [5]Glenn Laboratories for the Biology of Aging at Stanford, Stanford, United States; [6]Wu Tsai Neurosciences Institute, Stanford University, Stanford, United States

**Abstract** The African turquoise killifish is a powerful vertebrate system to study complex phenotypes at scale, including aging and age-related disease. Here, we develop a rapid and precise CRISPR/Cas9-mediated knock-in approach in the killifish. We show its efficient application to precisely insert fluorescent reporters of different sizes at various genomic loci in order to drive cell-type- and tissue-specific expression. This knock-in method should allow the establishment of humanized disease models and the development of cell-type-specific molecular probes for studying complex vertebrate biology.

## Editor's evaluation

This paper describes a rapid and easy to implement CRISPR/Cas9-mediated knock-in approach to precisely insert large transgenes in the African turquoise killifish. The methodologies performed are rigorous and the conclusions reached are well supported by the data. The established method is instrumental for many researchers working with unusual model species, and, in particular will expand the killifish community toolbox. It will revolutionize the field and bring the killifish, an emerging animal model in aging biology and disease modeling in vertebrates, into the spotlight even more.

*For correspondence:
abrunet1@stanford.edu

†These authors contributed equally to this work

Competing interest: The authors declare that no competing interests exist.

## Introduction

Studying complex biological phenotypes such as aging and disease in vertebrates is limited by issues of scale and speed. For example, the inherent long lifespan and low-throughput nature of mice prohibit iterative genetics and exploration of vertebrate biology. The African turquoise killifish *Nothobranchius furzeri* (hereafter killifish) has emerged as a powerful model to overcome this challenge and accelerate discovery due to its rapid timeline for sexual maturity (3–4 weeks post hatching) and naturally compressed lifespan (4–6 months) (*Hu and Brunet, 2018*; *Kim et al., 2016*). The killifish has the shortest generation time of a vertebrate model system bred in the laboratory (2 months) (*Hu and Brunet, 2018*; *Kim et al., 2016*; *Polačik et al., 2016*), making rapid vertebrate genetics possible. Tools to advance genetic interrogation of the killifish have been developed, including a sequenced genome (*Reichwald et al., 2015*; *Valenzano et al., 2015*), Tol2 transgenesis (*Allard et al., 2013*; *Hartmann and Englert, 2012*; *Valenzano et al., 2011*), CRISPR/Cas9-mediated knock-out (*Harel et al., 2015*), and CRISPR/Cas13-mediated knock-down (*Kushawah et al., 2020*). This genetic toolkit has enabled discoveries about the mechanisms of aging (*Astre et al., 2022a*; *Bradshaw et al., 2022*; *Chen et al., 2022*; *Harel et al., 2022*; *Louka et al., 2022*; *Matsui et al., 2019*; *Smith et al., 2017*; *Van*

*Houcke et al., 2021b*; *Vanhunsel et al., 2021*), regeneration (*Vanhunsel et al., 2022a*; *Vanhunsel et al., 2022b*; *Wang et al., 2020*), evolution (*Cui et al., 2019*; *Sahm et al., 2017*; *Singh et al., 2021*; *Willemsen et al., 2020*), development (*Abitua et al., 2021*; *Dolfi et al., 2019*), and embryonic diapause – a state of 'suspended animation' (*Hu et al., 2020*; *Singh et al., 2021*).

Knock-in methods are essential for the genetic tractability of model organisms. They enable precise mutations in key genes for mechanistic studies and human disease modeling. Knock-in technologies also allow the insertion of molecular tags or reporters at specific genomic loci. Combined with self-cleaving peptides, a knock-in approach can be leveraged to drive cell-type- and tissue-specific expression of ectopic genes (e.g. genes of interest, recombinases) or probes (e.g. fluorescent reporters, calcium indicators). Thus, developing a method to precisely insert large genes and allow the efficient generation of stable lines with germline transmission is critical for establishing the killifish as a system for genetic engineering at scale.

## Results
### CRISPR/Cas9-mediated knock-in in killifish allows efficient tissue-specific expression of fluorescent reporters

To achieve precise integration of genes of interest at endogenous target loci, we developed a method based on CRISPR/Cas9-mediated homology-directed repair (HDR). CRISPR/Cas9-mediated HDR is often associated with issues of low efficiency and multicopy insertion (*Auer et al., 2014*). To overcome these issues, we injected killifish embryos with a cocktail composed of (1) recombinant Cas9 protein, (2) synthetic guide RNAs (gRNAs), (3) a chemically-modified linear double-stranded DNA (dsDNA) HDR template, and (4) a small molecule HDR enhancer which inhibits non-homologous end joining (NHEJ) from Integrated DNA Technologies (IDT) (*DiNapoli et al., 2020*) (see Methods; *Supplementary file 1*). We designed the dsDNA HDR template with 150–200 bp homology arms flanking the site of insertion at the target locus (in this case, the stop codon of a specific gene) (*Figure 1A*; *Supplementary file 2*). The length of homology arms was selected based on IDT's recommendations for longer (>200 bp) insertions (see Materials and methods). The target site for genomic insertion was designed within 2–7 bp of the Cas9 cut sites (*Supplementary file 3*). To rapidly assess the efficiency of CRISPR/Cas9-mediated knock-in, we included the following sequences in the dsDNA HDR template: a *T2A* sequence [encoding the T2A self-cleaving peptide (*Szymczak et al., 2004*)] and the fluorescent protein Venus (*Nagai et al., 2002*). Use of the T2A self-cleaving peptide avoids direct fusion of the fluorescent protein to the targeted gene's protein product (*Figure 1A*; *Supplementary file 3*). The modified dsDNA HDR template and gRNAs can all be directly ordered (see Materials and methods), which alleviates the need for cloning, PCR, or *in vitro* transcription. With successful insertion, the expression of Venus should be controlled by the endogenous regulatory elements (e.g. promoter, enhancers) of the target gene, which could be leveraged for cell-type- or tissue-specific expression.

Using this approach, we targeted Venus to three distinct genomic loci in the killifish: *ELAVL3* (which encodes the HuC protein), *CRYAA* (which encodes a crystallin protein), and *ACTB2* (which encodes an actin protein). These three loci are known to have brain-specific (*Ahrens et al., 2012*), lens-specific (*Posner et al., 2017*), and ubiquitous (*Gutierrez-Triana et al., 2018*) expression, respectively, in teleost fish (including zebrafish and medaka). After injection of CRISPR/Cas9 reagents into one-cell stage killifish embryos, we waited 14–21 days for the embryos to develop and visually screened embryos for Venus fluorescence – indicative of protein expression and suggestive of successful CRISPR/Cas9-mediated knock-in. We observed Venus fluorescent protein expression in the expected tissues: developing brain for *ELAVL3*-targeted embryos (*Figure 1B*), lens of the eye for *CRYAA*-targeted embryos (*Figure 1B*; *Figure 1—figure supplement 1*), and in all cells of the embryo for *ACTB2*-targeted embryos (*Figure 1B*). In all embryos screened, we did not observe Venus expression in a tissue that was not specifically targeted.

For all three targeted loci, we observed Venus fluorescence (suggestive of successful CRISPR/Cas9-mediated knock-in) in over 40% of developed embryos (*Figure 1C*). For the *ELAVL3* locus, we achieved the highest CRISPR/Cas9-mediated knock-in efficiency using both a chemically modified dsDNA HDR template (i.e. IDT's Alt-R HDR Donor Blocks, see Methods) and a small molecule HDR enhancer as compared to an unmodified dsDNA HDR template (i.e. IDT's gBlocks, see Methods) and no small molecule HDR enhancer (*Figure 1C*; *Figure 1—figure supplement 2A*; *Supplementary file 4*). We

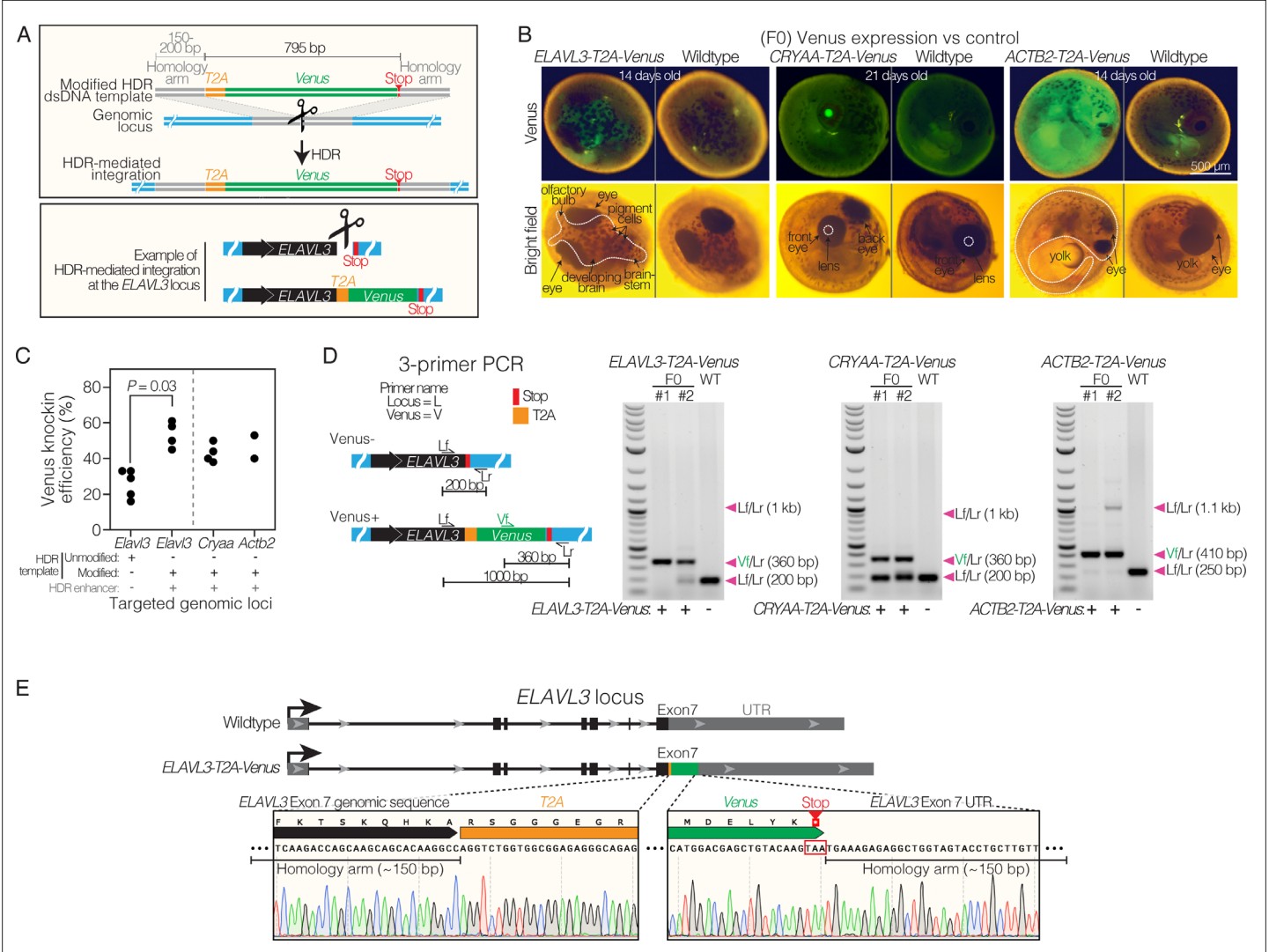

**Figure 1.** Efficient homology directed repair for precise knock-in at different genomic locations in killifish. (**A**) Schematic of *T2A-Venus* insertion at the *ELAVL3* locus. (**B**) Images of F0 Venus positive and wildtype 14–21-day-old embryos for each targeted locus (*ELAVL3*, *CRYAA*, and *ACTB2*). Twenty-one-day-old embryos were dried on coconut fiber for 7 days prior to imaging and have altered autofluorescence compared to 14-day-old embryos that were not yet put on coconut fiber. See *Figure 1—figure supplement 1*. (**C**) Efficiency of *T2A-Venus* knock-in at each locus (determined by visual inspection of Venus fluorescence in developed embryos) and efficiency of knock-in at *ELAVL3* with a dsDNA HDR template lacking chemical modification and without the small molecule HDR enhancer; 2–5 independent injection replicates per condition; n=32–249 injected embryos per replicate. *P*-value calculated using a two-tailed Mann-Whitney test. See *Figure 1—figure supplement 2*. Raw data in *Supplementary file 4*. (**D**) Left, 3-primer PCR schematic showing locus-specific external primers forward (Lf) and reverse (Lr) and internal forward Venus primer (Vf). Right, gel images of 3-primer PCR for each locus comparing F0 with wildtype (WT) fish. Arrowheads indicate each primer pair and its expected amplification product length. Scoring Venus positive (+) or negative (-) for each fish is indicated below the gel images. Note that the relatively large ~1 kb Lf/Lr product in the transgenic F0 animals is likely to be outcompeted by the shorter Vf/Lr amplification product during the PCR reaction. Raw gel image in *Figure 1—source data 1* and *Figure 1—source data 2*. (**E**) Top, comparison of *ELAVL3* locus for wildtype and *ELAVL3-T2A-Venus*. Bottom, precise in-frame insertion of *T2A-Venus* in exon 7, immediately before the stop codon of *ELAVL3* and followed by the *ELAVL3* untranslated region (UTR).

The online version of this article includes the following source data and figure supplement(s) for figure 1:

**Source data 1.** Original unedited gel shown in panel D.

**Source data 2.** Original unedited gel with relevant bands labeled shown in panel D.

**Figure supplement 1.** Venus expression in F0 *CRYAA-T2A-Venus* transgenic animals.

**Figure supplement 2.** Comparison of knock-in efficiency and embryo lethality using chemically modified dsDNA HDR templates and small molecule HDR enhancer.

**Figure supplement 2—source data 1.** Raw data for panel C.

therefore used the former approach for all subsequent constructs and loci. We did not observe significant differences in the lethality of embryos injected with CRISPR/Cas9 knock-in reagents (including chemically modified dsDNA HDR template and/or small molecule HDR enhancer) compared to non-injected wildtype embryos (*Figure 1—figure supplement 2B, C*). In general, the lethality of killifish embryos, whether injected or not, was variable in the first two weeks of development (dependent on breeders and clutch; *Figure 1—figure supplement 2B, C*). Thus, to ensure successful generation of transgenic animals that survive past development, we recommend injecting ~100 embryos per line.

Importantly, we confirmed that the genomic knock-in occurred at the expected genomic locus by PCR genotyping of DNA extracted from tail clips of the injected individuals (F0 founders) using 3 primers, 2 surrounding the insertion site for each gene and 1 in the *Venus* transgene (*Figure 1D*; *Supplementary file 3*; see below for sequencing confirmation in the F1 generation). Knock-in at the *ACTB2* locus resulted in Venus positive embryos that did not survive hatching, consistent with what has been observed for *ACTB2* knock-in in medaka (*Gutierrez-Triana et al., 2018*) and perhaps due to the sensitivity of actin assembly to any perturbation (e.g. additional amino acids due to cleavage of the P2A peptide). Thus, to genotype Venus positive embryos targeting the *ACTB2* locus, DNA was extracted from whole embryos (not tail clips). Genotyping injected animals using this 3-primer PCR strategy enables a rough estimate of the level of heterozygosity and mosaicism in each animal. This estimate is helpful in selecting highly edited F0 founders for generating stable lines, especially in cases where the introduced insertion does not contain a fluorescent reporter and thereby cannot be visually selected (*Figure 1D*). Hence, this CRISPR/Cas9-mediated knock-in method allows for precise and efficient editing at several loci, including tissue-specific ones; however, it may be more difficult to use this knock-in approach to generate stable lines targeting essential genes encoding proteins involved in sensitive assemblies such as *ACTB2*.

## Germline transmission of CRISPR/Cas9-mediated knock-in and generation of stable lines

A key aspect of genome editing is germline transmission to allow the generation of genetically modified lines. To determine if the CRISPR/Cas9-mediated insertion can be transmitted to the next generation, we evaluated the efficiency of germline transmission using transgenic *ELAVL3-T2A-Venus* founders (F0s). Sixty-seven percent of F0 founders, when crossed with wildtype fish, produced Venus-positive F1 progeny (*Figure 2A and B*). Given the high efficiency of germline transmission and the rapid generation time of killifish, we tested if it was possible to directly generate homozygous F1 animals by inter-crossing genetically modified F0 individuals (*Figure 2C and D*). As an example, we used one Venus-positive F0 male (F0 #1; *Figure 2B*) and one Venus-positive F0 female (F0 #2; *Figure 2B*), both of which produced a large fraction of Venus-positive progeny when crossed with wildtype fish (*Figure 2B*). Upon inter-crossing these founders, we found that 85% of the resulting F1 Venus-positive progeny were homozygous for the insertion at the *ELAVL3* locus (*Figure 2D*). PCR amplification and genotyping by Sanger sequencing of homozygous F1 animals confirmed that the *T2A-Venus* integration at the *ELAVL3* locus was as designed—single copy and in frame, with no observed mutations within 1 kb around the insertion site (*Figure 1E*; *Figure 2A, C and D*; *Figure 2—figure supplement 1*). This accelerated inter-crossing approach could enable rapid testing of homozygous F1 lines if desired. However, we also note that directly generating homozygous F1 lines may also increase the risk of propagating silent mutations from F0 animals (e.g. off-target genome editing in F0s), which might in turn lead to phenotypes independent of the introduced transgene. We formally tested for potential off-target insertions/mutations upon CRISPR/Cas9-mediated knock-in for *ELAVL3* targeting. PCR amplification and Sanger sequencing of homozygous F1 *ELAVL3-T2A-Venus* animals at the three most likely off-target sites [predicted by CHOPCHOP (*Labun et al., 2019*)] showed no off-target editing at these sites (*Figure 2—figure supplement 2*). Nevertheless, to limit the risk of off-target editing, we recommend backcrossing founders to wildtype animals.

To characterize the stable *ELAVL3-T2A-Venus* knock-in line, we examined the expression pattern of Venus at different stages. At the larval stage, *ELAVL3-T2A-Venus* F1 homozygous individuals exhibit specific and strong Venus expression throughout the nervous system including the retina, brain, and spinal cord (*Figure 2E*; anti-GFP staining), which is expected given that the *ELAVL3* promoter is commonly used as a pan-neuronal promoter in larval zebrafish (*Ahrens et al., 2012*). In adults, *ELAVL3-T2A-Venus* F3 heterozygous individuals show strong *Venus* expression in the brain (*Figure 2F*)

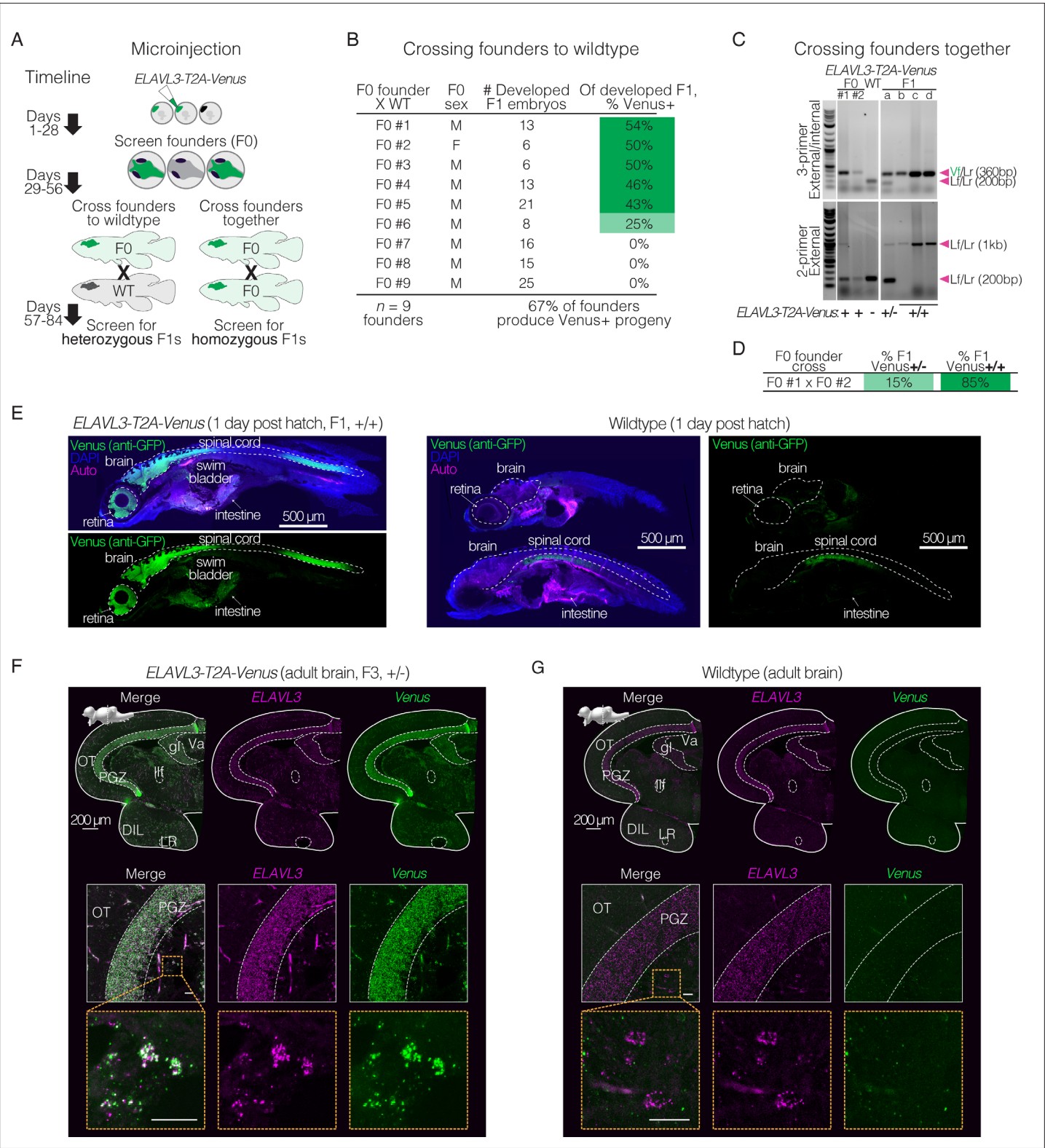

**Figure 2.** Rapid generation of stable knock-in lines in the killifish. (**A**) Schematic of generating a stable knock-in line by either crossing F0 X WT (left) or F0 X F0 (right), with timelines to verified heterozygous or homozygous animals. (**B**) Germline transmission of *T2A-Venus* at the *ELAVL3* locus was verified by crossing F0 X WT (n=9 breeding pairs), with 6–25 developed embryos per pair screened visually or by PCR. (**C**) Crossing F0 animals positive for *T2A-Venus* at the *ELAVL3* locus (F0 X F0). Gels showing F0 parents (left) and F1 progeny (a, b, c, and d; right) with 3-primer PCR (top) and external PCR (bottom) using Venus and locus-specific primers shown in *Figure 1D*. The external PCR shows both heterozygous (a) and homozygous (b, c, and d) F1

*Figure 2 continued*

progeny for *ELAVL3-T2A-Venus*. Arrowheads indicate each primer pair and its expected amplification product length. Scoring for each lane of the gel is indicated below the gel images. F0 animals are likely mosaic so only '+' or '-' was assigned based on the 3-primer PCR result. See *Figure 2—figure supplement 1* and *Figure 2—figure supplement 2*. Raw gel images in *Figure 2—source data 1*; *Figure 2—source data 2* and *Figure 2—source data 3*. (**D**) Percent of fully developed and Venus-positive F1 progeny from the F0 X F0 cross that are heterozygous (+/-) or homozygous (+/+) for insertion of *Venus* in *ELAVL3-T2A-Venus* animals. (**E**) Immunofluorescence of sagittal sections of larval (1 day post hatch) F1 homozygous *ELAVL3-T2A-Venus* (left) compared with wildtype (right) killifish showing merged images of Venus (stained with anti-GFP antibody; green), DAPI (nuclei; blue), and autofluorescence ('Auto'; magenta) as well as separate images from different channels for Venus (stained with anti-GFP antibody; green). In both *ELAVL3-T2A-Venus* and wildtype individuals, we observe background green signal for example in the intestine and ventral to the spinal cord. This background signal is not identical between *ELAVL3-T2A-Venus* and wildtype samples, likely due to slight differences in depth of the sagittal slice. Scale bar = 500 µm. (**F**) Top row, *in situ* hybridization (via HCR) of coronal brain section of an adult (3 months old) F3 heterozygous *ELAVL3-T2A-Venus* male showing merged images of *Venus* transcript (green) and the *ELAVL3* transcript (magenta) as well as images from separate channels. Scale bar = 200 µm. Upper left corner, sagittal view of the killifish brain indicating the plane of the coronal section. Middle row, zoom in on the periglomerular gray zone (PGZ) of the optic tectum (OT). Bottom row, zoom in on individual *ELAVL3*-expressing cells. While *ELAVL3* (magenta) and *Venus* (green) signal largely colocalize, there is some non-colocalizing background signal. This background is similar to signal observed in wildtype animals (see **G**). Scale bar = 20 µm. See *Figure 2—figure supplement 3*. (**G**) Top row, *in situ* hybridization (via HCR) of coronal brain section of an adult (3 months old) wildtype male showing merged images of *Venus* transcript (green) and the *ELAVL3* transcript (magenta) as well as images from separate channels. Scale bar = 200 µm. Upper left corner, sagittal view of the killifish brain indicating the plane of the coronal section. Middle row, zoom in on the periglomerular gray zone (PGZ) of the optic tectum (OT). Bottom row, zoom in on individual *ELAVL3*-expressing cells. We observe some background green signal in wildtype animals; however, overall green signal is much less than what is observed in the *ELAVL3-T2A-Venus* animals (**F**). Scale bar = 20 µm.

The online version of this article includes the following source data and figure supplement(s) for figure 2:

**Source data 1.** Original unedited gel shown in panel C.

**Source data 2.** Original unedited gel shown in panel C.

**Source data 3.** Original unedited gel with relevant bands labeled shown in panel C.

**Figure supplement 1.** PCR amplification of F0 parents and F1 progeny confirms *T2A-Venus* integration and germline transmission at the *ELAVL3* locus.

**Figure supplement 1—source data 1.** Original unedited gel shown in panel B.

**Figure supplement 1—source data 2.** Original unedited gel with relevant bands labeled shown in panel B.

**Figure supplement 1—source data 3.** Original unedited gel shown in panel C.

**Figure supplement 1—source data 4.** Original unedited gel with relevant bands labeled shown in panel C.

**Figure supplement 2.** Evaluation of potential off-target effects in homozygous F1 CRISPR/Cas9 knock-in fish.

**Figure supplement 3.** Assessment of *Venus* and *ELAVL3* transcript levels in *ELAVL3-T2A-Venus* knock-in and wildtype control.

**Figure supplement 3—source data 1.** Raw data for panels B and C.

(while we did not specifically test retina and spinal cord in adults, Venus expression should also occur in these areas). Within the adult brain, many brain regions and cells showed strong Venus expression (*Figure 2F*). *In situ* hybridization indicates that the *Venus* transcript indeed colocalizes to the same brain regions and cells as the *ELAVL3* transcript (*Figure 2F*; *Supplementary file 5*). The *ELAVL3* transcript expression pattern in the *ELAVL3-T2A-Venus* line is consistent with that of *ELAVL3* in wildtype animals (*Figure 2G*). Together, these results suggest that the knocked-in *Venus* transgene recapitulates the endogenous expression pattern of the targeted *ELAVL3* locus in the brain.

The insertion of sequences into the genome could inadvertently impair endogenous expression of the targeted locus. To test this possibility in a relatively quantitative manner at the *ELAVL3* locus, we performed reverse-transcription followed by quantitative PCR (RT-qPCR) to compare transcripts in wildtype, heterozygous, and homozygous siblings for the *ELAVL3-T2A-Venus* knock-in allele (*Figure 2—figure supplement 3A*). As expected, we observed a large increase in expression of *Venus* in animals heterozygous or homozygous for the *ELAVL3-T2A-Venus* allele compared to wildtype animals (with highest expression of *Venus* in homozygous animals; *Figure 2—figure supplement 3B*). *ELAVL3* expression level was not significantly different between animals heterozygous or homozygous for the *ELAVL3-T2A-Venus* allele compared to wildtype siblings (*Figure 2—figure supplement 3C*; although there might be a slight, non-significant reduction in the homozygous mutants). Thus, inserting a fluorescent protein gene at the endogenous *ELAVL3* locus does not significantly impair the expression of the endogenous gene.

Collectively, these observations indicate that the knock-in method we developed enables generation of stable lines of transgenic vertebrate animals in 2–3 months.

# Insertion of long sequences into the genome to drive gene expression in a cell- or tissue-specific manner

We asked if this CRISPR/Cas9-mediated insertion method could be used to insert longer sequences into the killifish genome for expression in specific cells or tissues. The insertion of long sequences at a precise genomic location, while technically challenging, is critical for leveraging the cell-type or tissue specificity of a particular locus to drive ectopic expression of specific genes or molecular probes. We designed a longer dsDNA HDR template that would result in a 2 kb long insertion sequence. This HDR template includes two consecutive fluorescent proteins (Venus and oScarlet) targeted to the *ELAVL3* locus, with *T2A* and *P2A* sequences (encoding another self-cleaving peptide) 5' to each fluorescent protein, respectively, to avoid direct fusion. The oScarlet was also tagged with the nuclear localization

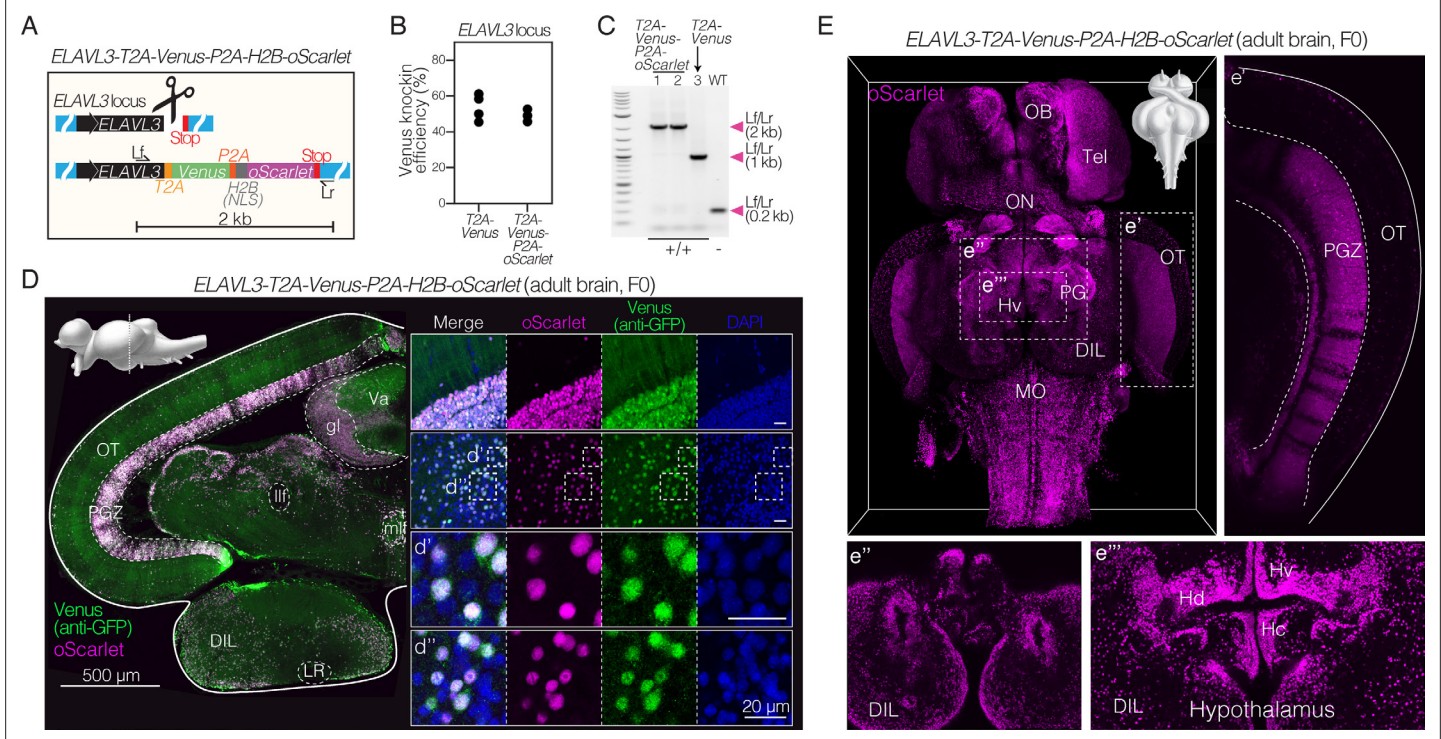

**Figure 3.** Efficient and stable knock-in of a large 2 kb insertion in killifish. (**A**) Schematic of design of a *T2A-Venus-P2A-H2B-oScarlet* sequence for targeted knock-in at the *ELAVL3* locus and locus-specific external primers forward (Lf) and reverse (Lr). (**B**) Knock-in efficiency comparing 2 kb insertion (*ELAVL3-T2A-Venus-P2A-H2B-oScarlet*) to the 0.8 kb insertion (*ELAVL3-T2A-Venus*) determined by visual inspection of developed F0 embryos for Venus fluorescence; 3–4 independent injection replicates per condition; n=80–157 embryos injected per replicate. No significant difference between groups (two-tailed Mann-Whitney test; p-value = 0.6). Data in *Supplementary file 4*. (**C**) PCR amplification at the *ELAVL3* locus using locus-specific external primers forward (Lf) and reverse (Lr) shown in (**A**) comparing amplicon length from two F1 *ELAVL3-T2A-Venus-P2A-H2B-oScarlet* animals (lane 1 and 2), one F1 *ELAVL3-T2A-Venus* animal (lane 3), and one wildtype animal (lane WT), showing a single band at the expected length in each case. Scoring for each lane of the gel is indicated below the gel image. See *Figure 3—figure supplement 1*. Raw gel image in *Figure 3—source data 1* and *Figure 3—source data 2*. (**D**) Left, immunofluorescence of coronal brain section of adult (3 months old) F0 *ELAVL3-T2A-Venus-P2A-H2B-oScarlet* male, showing expression of Venus and oScarlet. Scale bar = 500 μm. Upper left corner, sagittal view of the killifish brain indicating the plane of the coronal section. Right, select regions showing separate channels for oScarlet (magenta), Venus (stained with anti-GFP antibody; green), DAPI (nuclei; blue) as well as merged channels. (**d'**) and (**d''**): zoomed in individual cells. Scale bar = 20 μm. oScarlet expression is confined to nuclei while Venus expression is observed throughout cell bodies and projections. The DAPI positive, oScarlet/Venus negative cells are likely non-neuronal brain cell types (e.g., astrocytes, oligodendrocytes, etc.). (**E**) Brain-wide expression of nuclear-localized oScarlet (magenta) in adult (1 month old) F0 *ELAVL3-T2A-Venus-P2A-H2B-oScarlet* male. Select regions are highlighted: (**e'**) the optic tectum (OT), (**e''**) the most ventral view of the hypothalamus, and (**e'''**) the periventricular hypothalamus. Strips in the PGZ that appear to lack oScarlet expression could be due to mosaicism (i.e. lack of knock-in in some cells) in the F0 individual.

The online version of this article includes the following source data and figure supplement(s) for figure 3:

**Source data 1.** Original unedited gel shown in panel C.

**Source data 2.** Original unedited gel with relevant bands labeled shown in panel C.

**Figure supplement 1.** Evaluation of potential off-target effects in homozygous F1 CRISPR/Cas9 knock-in fish.

signal (NLS) from human histone 2B, a NLS commonly used for zebrafish transgenics (*Freeman et al., 2014*; *Kanda et al., 1998*), to allow nuclear localization of this fluorescent protein. The resulting insertion sequence is 2 kb long – a length that would encode proteins of ~670 amino acids and ~74 kDa (*Figure 3A*; *Supplementary file 3*; *Supplementary file 6*). We observed successful CRISPR/Cas9-mediated knock-in of this longer sequence in ~50% of developed embryos (*Figure 3B*). There was no decrease in efficiency or change in embryo lethality for this longer insertion relative to the shorter (0.8 kb) insertion previously tested at the same locus (*Figure 3B*; *Figure 1—figure supplement 2B, C*). PCR amplification and genotyping by Sanger sequencing of homozygous F1 animals confirmed that the *T2A-Venus-P2A-H2B-oScarlet* integration at the *ELAVL3* locus was the expected size and in frame without mutations (*Figure 3C*; *Supplementary file 3*). PCR amplification and Sanger sequencing of homozygous F1 animals at the three predicted most likely off-target sites showed no off-target editing in these fish (*Figure 3—figure supplement 1*). Imaging coronal brain sections of adult F0 *ELAVL3-T2A-Venus-P2A-H2B-oScarlet* killifish showed cells expressing both oScarlet and Venus (*Figure 3D*). As expected, oScarlet expression was confined to nuclei while Venus expression was seen in both cell bodies and projections (*Figure 3D*). Imaging the whole brain of adult *ELAVL3-T2A-Venus-P2A-H2B-oScarlet* killifish revealed oScarlet-positive nuclei throughout the brain (*Figure 3E*). Thus, this method allows for pan-neuronal expression in the adult brain and could be leveraged to drive expression of molecular tools e.g. the optogenetic ion channel channelrhodopsin [~1 kb] (*Boyden et al., 2005*) or the genetically encoded calcium indicator GCaMP [~1.3 kb] (*Ahrens et al., 2012*) in a neuronal-specific manner.

## Cell-type-specific expression in subsets of neurons by targeting neuropeptide loci

We determined if this CRISPR/Cas9-mediated insertion could be used to build killifish reporter lines for specific cell types, notably neuronal subpopulations. This development is critical for systems neuroscience, including circuit-based studies. We focused on targeting neurons expressing neuropeptide Y (NPY) and hypocretin (HCRT). These neuronal populations are critical for organismal homeostasis through modulation of behaviors, including feeding behavior (*Jeong et al., 2018*) and sleep-wake behavior (*Chiu and Prober, 2013*; *Prober et al., 2006*; *Singh et al., 2017*). Growing evidence also suggests that these neuronal populations may be altered with age (*Fronczek et al., 2012*; *Hunt et al., 2015*; *Li et al., 2022*; *Montesano et al., 2019*).

We designed a dsDNA HDR template encoding Venus targeting the *NPY* or *HCRT* locus and with a *T2A* sequence 5' to the fluorescent protein sequence to avoid direct fusion between the neuropeptide and the fluorescent protein (*Figure 4A*; *Supplementary file 3*; *Supplementary file 7* and *Supplementary file 8*). PCR amplification and genotyping by Sanger sequencing of F1 animals confirmed that the *T2A-Venus* integration at the *NPY* or *HCRT* locus was as designed—single-copy and in frame at the targeted genomic location (*Figure 4A*; *Figure 4—figure supplement 1*; *Supplementary file 3*). Imaging of coronal brain sections of the adult *HCRT-T2A-Venus* killifish line showed a dense and isolated population of Venus-positive cell bodies in the dorsal periventricular hypothalamus (Hd; homologous to the mammalian arcuate nucleus; *Figure 4B*), consistent with endogenous *HCRT* expression previously reported in the adult killifish brain (*D'angelo, 2013*; *Montesano et al., 2019*). In contrast, the adult *NPY-T2A-Venus* line exhibited Venus-positive cell bodies throughout the brain, including in the central posterior thalamic nucleus (CP), anterior tuberal nucleus (TNa), periventricular and lateral hypothalamus, as well as in the periventricular gray zone (PGZ) of the optic tectum (OT) (*Figure 4D*), consistent with endogenous *NPY* expression previously reported in the adult killifish brain (*D'angelo, 2013*; *Montesano et al., 2019*).

Using *in situ* hybridization, we verified that *Venus* transcript indeed co-localized with the *HCRT* transcript in the *HCRT-T2A-Venus* line (*Figure 4C*; *Supplementary file 5*) or the *NPY* transcript in the *NPY-T2A-Venus* line (*Figure 4E*; *Supplementary file 5*). The *Venus* expression profiles observed in the *NPY-T2A-Venus* and *HCRT-T2A-Venus* lines were consistent with those of endogenous *NPY* and *HCRT* transcripts, respectively, in wildtype animals (*Figure 4—figure supplement 2*; *Supplementary file 5*).

To further validate the stable *NPY-T2A-Venus* knock-in line, we determined whether the expression level of the *NPY* transcript is impacted by the presence of the transgene. We performed RT-qPCR analysis comparing *NPY* transcripts in wildtype, heterozygous, and homozygous siblings for the *NPY-T2A-Venus* knock-in allele. As expected, we observed a large increase in expression of *Venus*

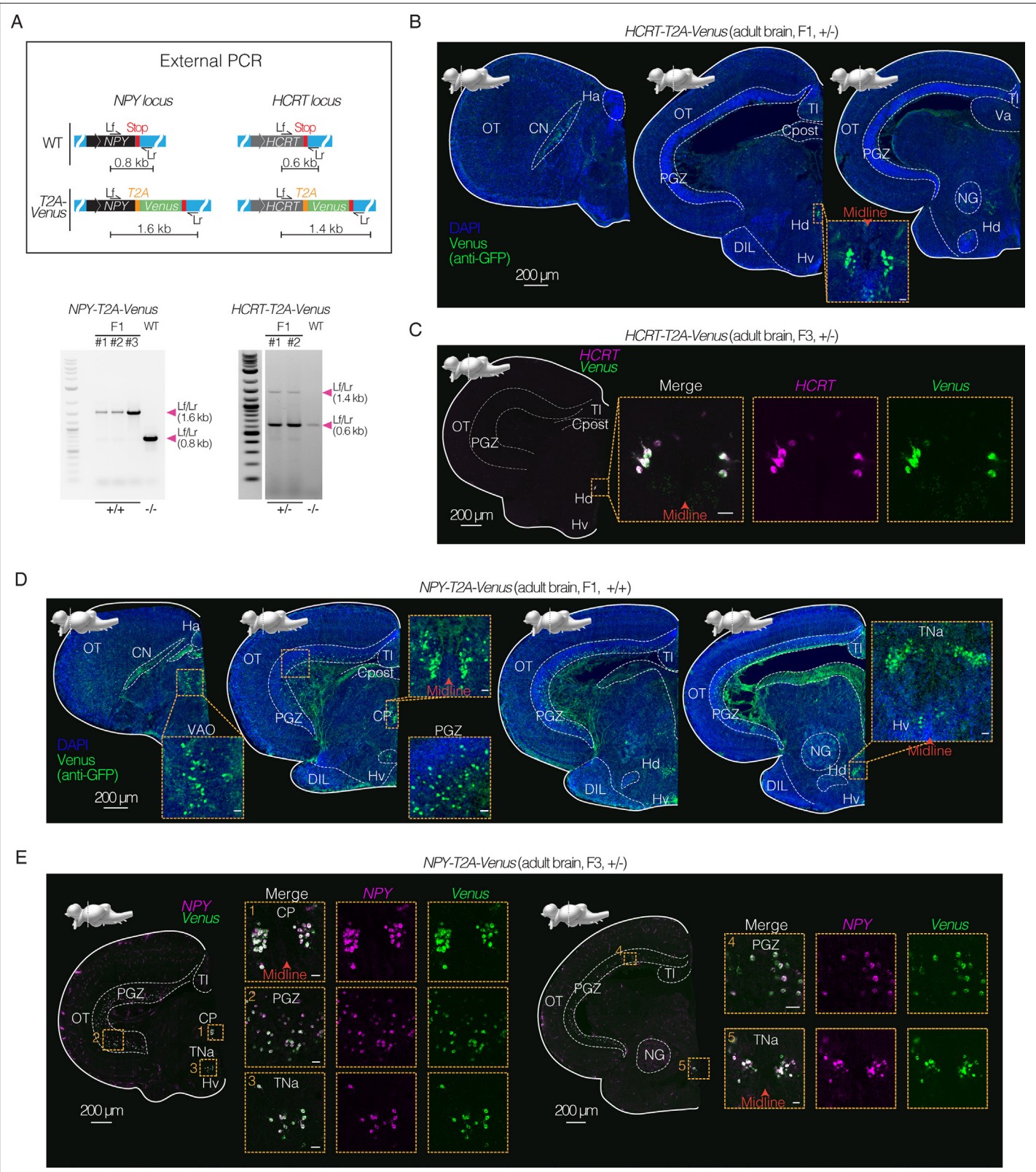

**Figure 4.** Expression in specific neuronal populations using CRISPR/Cas9 knock-in lines in killifish. (**A**) Top, schematics of design of *T2A-Venus* sequence for targeted knock-in at the *NPY* and *HCRT* loci including locus-specific external primers forward (Lf) and reverse (Lr). Bottom, PCR amplification at the *NPY* or *HCRT* locus comparing amplicon length from *NPY-T2A-Venus* (F1 animals) versus wildtype (WT) and comparing amplicon length from *HCRT-T2A-Venus* (F1 animals) versus wildtype (WT). See *Figure 4—figure supplement 1*. Raw gel images in *Figure 4—source data 1*; *Figure 4—source*

*Figure 4 continued on next page*

*Figure 4 continued*

*data 2* and *Figure 4—source data 3*. (**B**) Immunofluorescence of coronal brain sections of adult (4 months old) F1 heterozygous *HCRT-T2A-Venus* female, showing Venus expression (stained with anti-GFP antibody; green) and DAPI (nuclei; blue). Scale bar = 200 µm. Distinct nuclei indicated and labeled with abbreviated names. Above each slice is the sagittal view of the killifish brain indicating the plane of the coronal section. Inset shows zoom in on Venus positive population of cells in the dorsal hypothalamus close to the midline. Scale bar = 20 µm. See *Figure 4—figure supplement 2*. (**C**) *In situ* hybridization (via HCR) of coronal brain section of adult (4 months old) F3 heterozygous *HCRT-T2A-Venus* male, showing merged images of *Venus* transcript (green) and *HCRT* transcript (magenta). Scale bar = 200 µm. Inset shows zoom in on *Venus* positive and *HCRT*-positive population of cells in the dorsal hypothalamus close to the midline with merge and separated channels. Scale bar = 20 µm. See *Figure 4—figure supplement 2*. (**D**) Immunofluorescence of coronal brain sections of adult (3.5 months old) F1 homozygous *NPY-T2A-Venus* male, showing Venus expression (stained with anti-GFP antibody; green) and DAPI (nuclei; blue). Scale bar = 200 µm. Above each slice is the sagittal view of the killifish brain indicating the plane of the coronal section. Insets show zoom in on the Venus-positive populations. Scale bar = 20 µm. See *Figure 4—figure supplement 2*. (**E**) *In situ* hybridization (via HCR) of coronal brain sections of adult (3 months old) F3 heterozygous *NPY-T2A-Venus* male, showing merged image of *Venus* transcript (green) and the *NPY* transcript (magenta) *in situ* hybridization. Scale bar = 200 µm. Insets show zoom in on *Venus*-positive and *NPY*-positive populations. Scale bar = 20 µm. See *Figure 4—figure supplement 2* and *Figure 4—figure supplement 3*.

The online version of this article includes the following source data and figure supplement(s) for figure 4:

**Source data 1.** Original unedited gel shown in panel A.

**Source data 2.** Original unedited gel shown in panel A.

**Source data 3.** Original unedited gel with relevant bands labeled shown in panel A.

**Figure supplement 1.** Confirmation of knock-in by PCR amplification and sequencing.

**Figure supplement 1—source data 1.** Original gel.

**Figure supplement 1—source data 2.** Original unedited gel with relevant bands labeled.

**Figure supplement 2.** Staining of endogenous *NPY* and *HCRT* transcripts in the adult brain of wildtype killifish.

**Figure supplement 3.** Assessment of *Venus* and *NPY* transcript levels in *NPY-T2A-Venus* knock-in and wildtype control.

**Figure supplement 3—source data 1.** Raw data for panels B and C.

in animals heterozygous or homozygous for the *NPY-T2A-Venus* allele compared to wildtype animals (with highest expression of *Venus* in homozygous animals; *Figure 4—figure supplement 3*). In animals heterozygous for the *NPY-T2A-Venus* allele, *NPY* expression level was similar to that of wild-type animals (*Figure 4—figure supplement 3*). However, in animals homozygous for the *NPY-T2A-Venus* allele, *NPY* expression level was slightly but significantly lower compared to wildtype siblings (*Figure 4—figure supplement 3*). These observations suggest that the presence of the transgene at homozygous levels slightly impairs expression of endogenous *NPY*. Thus, in this case (and likely in other cases), use of heterozygous animals with edited alleles may be more conservative to preserve robust expression of the endogenous locus. The observation that the expression pattern of Venus recapitulates that of the targeted endogenous gene for all genes targeted here also supports the notion that in-frame off-target insertions are rare with this method. The generation of these lines serves as proof of principle that CRISPR/Cas9-mediated knock-in is a powerful method in killifish to drive cell-type-specific expression. These neuron-specific lines should also help the development of the killifish for systems neuroscience studies.

## Discussion

Here, we establish an efficient and versatile method for rapid and precise genome engineering of the short-lived African turquoise killifish. This CRISPR/Cas9-mediated knock-in method can be leveraged for cell-type- and tissue-specific expression of ectopic genes and reporters to study complex pheno-types at scale. We observe CRISPR/Cas9-mediated knock-in of large inserts at five distinct genomic loci (*ELAVL3*, *CRYAA*, *ACTB2*, *NPY*, and *HCRT*), with >40% efficiency at tested loci (*ELAVL3*, *CRYAA*, and *ACTB2*). With this CRISPR/Cas9-mediated knock-in approach, we observe a >65% germline trans-mission rate for knock-in at the *ELAVL3* locus. CRISPR/Cas9-mediated knock-in efficiency and germline transmission rates may vary depending on the genomic locus being targeted (*Hsu et al., 2013*; *Labun et al., 2019*), which might be due to the chromatin accessibility of that locus. We have generated four stable knock-in lines in this work: *ELAVL3-T2A-Venus*, *ELAVL3-T2A-Venus-P2A-H2B-oScarlet*, *NPY-T2A-Venus*, and *HCRT-T2A-Venus*. The high efficiency of germline transmission we observe in killifish may be due to the relatively slow rate of early cell division after fertilization in the African

turquoise killifish (~4 times slower in this species relative to non-annual teleost fishes; *Dolfi et al., 2014*). The killifish model, with hundreds of embryos produced at a given time (for example using harem breeding), allows for easy and high-throughput injection of genome-editing machinery into embryos (*Harel et al., 2015*; *Hu and Brunet, 2018*; *Kim et al., 2016*; *Polačik et al., 2016*). Moreover, the killifish has the shortest generation time of any vertebrate model bred in captivity (*Hu and Brunet, 2018*; *Kim et al., 2016*). The development of rapid and efficient knock-in establishes the killifish as a system for precise genetic engineering at scale, which has been challenging so far in vertebrates.

There has been promising progress in CRISPR/Cas9-mediated knock-in in killifish. Knock-in was first reported for short insertions (~8 bp; though germline transmission of these inserts was not achieved) (*Harel et al., 2015*). More recently, longer CRISPR/Cas9-mediated knock-in insertions (~1 kb) were achieved in killifish (in F0 individuals) with 18–30% efficiency, though germline transmission was not shown (*Oginuma et al., 2022*). Moreover, knock-in of a long (1.5 kb) insertion was achieved for a specific locus with ~11% efficiency and successful germline transmission (germline transmission rate not shown) (*Krug et al., 2023*). Our approach for CRISPR/Cas9-mediated knock-in in killifish allows long insertions (up to 2 kb), has >40% efficiency, and high germline transmission rates. The knock-in method developed here uses reagents that are commercially available, eliminating the need for cloning and PCR and making this method easy to adopt. Together, the steps described here could serve as a blueprint for knock-in approaches in other emerging model organisms.

Previous studies have developed different CRISPR/Cas9-mediated long insertion knock-in approaches not only in killifish but also in other teleost species, such as zebrafish and medaka (*Gutierrez-Triana et al., 2018*; *Krug et al., 2023*; *Oginuma et al., 2022*; *Seleit et al., 2021*; *Wierson et al., 2020*). Key points of difference between these approaches include (1) type of donor repair template – plasmid donor (with *in vivo* linearization) (*Oginuma et al., 2022*; *Wierson et al., 2020*), dsDNA PCR amplification product (*Gutierrez-Triana et al., 2018*; *Krug et al., 2023*; *Seleit et al., 2021*), or cloning-free synthetic dsDNA (as reported here), (2) use of chemical modification on linear dsDNA HDR repair template to prevent unwanted integration events (e.g. concatemerization) and boost HDR efficiency – biotin (*Gutierrez-Triana et al., 2018*; *Krug et al., 2023*; *Seleit et al., 2021*), or IDT's proprietary modification available with Alt-R HDR Donor Block (as reported here), and (3) length of the homology arms – short (24–40 bp; *Seleit et al., 2021*; *Wierson et al., 2020*), middle-range (150–200 bp; as reported here), or long (300–900 bp; *Gutierrez-Triana et al., 2018*; *Krug et al., 2023*). While it is difficult without side-by-side experiments to compare each knock-in approach, high insertion efficiencies (>40%) and germline transmission have been achieved with different methods, for example, in zebrafish using plasmid donors with *in vivo* linearization and short homology arms (*Wierson et al., 2020*), in medaka using dsDNA PCR product donors, biotin modification and short homology arms (*Seleit et al., 2021*), and (reported here) in killifish using synthetic dsDNA donors, IDT's proprietary modification (Alt-R HDR Donor Block) and middle-range length (150–200 bp) homology arms. Thus, a variety of approaches for CRISPR/Cas9-mediated knock-in have been successful in teleost fish, expanding the toolkit that can be used in these species.

Beyond knock-in efficiency, there are also advantages and limitations of our reported method. Our method uses synthetic, chemically-modified repair templates with ~150–200 bp homology arms, which has the advantage of being easily adoptable without expertise in molecular biology techniques (cloning or PCR). In addition, there is no preparation required for reagents (e.g. cloning or amplification with cleanup steps), which helps save hands-on time. Finally, the use of synthetic reagents that are sequenced and quality-controlled reduces the likelihood of mutations in the HDR template. However, the use of synthetic HDR templates does have limitations: (1) there is a limit on the total length (≤3000 bp) and complexity of dsDNA HDR templates that can be synthesized (e.g. sequence containing repetitive elements), (2) there is a wait time to receive the synthesized template, and (3) synthesis of large templates can be expensive. A major point of consideration in choosing a particular approach may be the number of lines being generated. If generating a few CRISPR/Cas9 knock-in lines, a purely synthetic approach would likely require the least amount of hands-on work and start-up time. In contrast, if one is planning to design many lines, a modular cloning/PCR based approach may be most cost effective.

The CRISPR/Cas9-mediated knock-in approach we developed should allow the establishment of versatile strategies to probe complex phenotypes, including development, 'suspended animation' (embryonic diapause), regeneration, aging, and age-related diseases. Given the potential of the

African killifish for modeling human aging (*Hu and Brunet, 2018*; *Kim et al., 2016*; *Van Houcke et al., 2021a*), this knock-in method should also allow the generation of human disease models that can be studied longitudinally, over an entire lifespan. For example, this CRISPR/Cas9-mediated knock-in could be used to introduce human neurodegenerative disease variants into conserved endogenous killifish loci (e.g. amyloid precursor protein [*APP*] for Alzheimer's disease) or to drive neurodegenerative disease variants using a pan-neuronal promoter such as *ELAVL3* (though we have not examined how the expression level of *ELAVL3* changes with age in this study). Human disease variant models in mice have been critical to understand disease mechanisms and treatment strategies (*Dawson et al., 2018*; *Fisher and Bannerman, 2019*; *Jankowsky and Zheng, 2017*). Generating human disease models that are scalable and integrate both genetics and age as risk factors has the potential to identify new strategies to treat these diseases.

This study highlights the power of knock-in, combined with self-cleaving peptides, to drive cell-type-specific expression of ectopic genes such as molecular reporters (e.g. fluorescent reporters, calcium indicators), recombinases (e.g. Cre), and optogenetic tools (e.g. light sensitive ion channels such as channelrhodopsin). The cell-type resolution of this genetic tool should open studies in a variety of fields, including systems neuroscience. Additional variations, such as the use of 'landing pads' (i.e. genetic loci that support stable long-term expression of transgenes such as *ROSA26* in mice; *Soriano, 1999*) and inducible promoters (either endogenous or ectopic; *Gossen and Bujard, 1992*; *Gossen et al., 1995*), could be further developed to complete this toolkit. Overall, this knock-in method should accelerate the use of the killifish as a scalable vertebrate model and allow discoveries in several fields, including regeneration, neuroscience, aging, and disease, with conserved implications for humans.

## Materials and methods

### African turquoise killifish care and husbandry

African turquoise killifish (GRZ strain) were maintained according to established guidelines (*Astre et al., 2022b*; *Bedbrook et al., 2023*; *Nath et al., 2023*; *Reichard et al., 2022*; *Žák et al., 2020*). Briefly, animals were housed at 26–27 °C in a central filtration recirculating system (Aquaneering, San Diego) at a conductivity between 3800–4000 µS/cm and a pH between 6.5–7.0, with a daily exchange of 10% water treated by reverse osmosis (i.e. RO water). Animals were kept on a 12 hr light/dark cycle and were fed twice a day on weekdays and once a day on weekends. Adult fish (>1 month of age) were fed dry fish food (Otohime fish diet, Reed Mariculture, Otohime C1) while young fish (<1 month of age) were fed freshly hatched brine shrimp (Brine Shrimp Direct, BSEP6LB). Killifish embryos were raised in Ringer's solution (Millipore, 96724), with two tablets per liter of RO water and 0.01% methylene blue (i.e. embryo solution) at 26–27 °C in 60 mm x 15 mm petri dishes (Fisher Scientific, 07-000-328) at a density of <100 embryos per plate. After two weeks in embryo solution, embryos were transferred to moist autoclaved coconut fiber (Eco Earth Coconut Fiber, EE-8) lightly packed in petri dishes where they were incubated for another two weeks at 26–27 °C. After 2–3 weeks on moist coconut fiber, embryos were hatched. For hatching, embryos were placed in humic acid solution (1 g/l, Sigma-Aldrich, 53680 in RO water) and incubated overnight at room temperature. While we did not specifically track fertility rates of the four stable knock-in lines generated in the paper (*ELAVL3-T2A-Venus*, *ELAVL3-T2A-Venus-P2A-H2B-oScarlet*, *NPY-T2A-Venus*, and *HCRT-T2A-Venus*), we did not observe obvious differences in fertility, and we did not struggle to maintain these lines. All animals were raised in accordance with protocols approved by the Stanford Administrative Panel on Laboratory Animal Care (protocol #APLAC-13645).

### Design of guide RNA sequences

For each selected gene, gRNA target sites were identified using CHOPCHOP (*Labun et al., 2019*) (https://chopchop.rc.fas.harvard.edu/) with the Nfu_20140520/Jena genome. One guide sequence was selected for each target gene of interest. Guide sequences were only selected if followed by the PAM site (5'-NGG-3') for *Streptococcus pyogenes* Cas9. The necessity of a PAM site near the target insertion site is a constraint of a CRISPR/Cas9-mediated knock-in approach. The Cas9 cut sites were between 2 and 7 bp from the target insertion site (*Supplementary file 3*). Guide RNAs were designed for compatibility with Integrated DNA Technologies' (IDT, Coralville, IA) Alt-R method. For detailed

methods and design tools, see https://sg.idtdna.com/pages. All Alt-R CRISPR RNAs (crRNAs) and universal trans-activating crRNA (tracrRNA) were chemically synthesized (2 nmol, IDT). Synthetic Alt-R crRNA and tracrRNA were resuspended in nuclease-free duplex buffer (IDT) to a final concentration of 100 µM each and stored at –20 °C. The guide sequences of all crRNAs used in this study are provided in *Supplementary file 3*. In this work, we used the two-part synthetic tracrRNA and crRNA purchased from IDT to generate a functional gRNA. This system is more cost-effective than purchasing the full length synthetic single guide RNA (sgRNA). Another cost-effective option is to perform in-house synthesis of sgRNA via *in vitro* transcription (e.g. MEGAscript T7 Transcription Kit), although this requires more hands-on time. While sgRNAs have been used and validated in killifish for CRISPR/Cas9-mediated engineering (*Harel et al., 2015*; *Harel et al., 2016*), we did not evaluate sgRNAs for CRISPR/Cas9-mediated knock-in in this study.

## Design of DNA templates for HDR

Double-stranded DNA (dsDNA) HDR templates were designed with 150–200 bp homology arms containing DNA sequences surrounding the target Cas9 cut site. The length of homology arms was selected based on IDT's recommendations for the Alt-R HDR Donor Block product, which is the dsDNA donor used for most experiments in the paper (see below). IDT reports that 100–300bp homology arms resulted in efficient knock-in for insertions between 0.5 and 2kb for K562 cells in culture. We selected a length on the shorter end of that scale such that the dsDNA HDR template was less expensive for synthesis. Homology arms began within 2–7 bp of the Cas9 cut site. HDR template sequences used in this study are provided in *Supplementary file 3* and fully annotated designs are provided in *Supplementary file 2*; *Supplementary file 6*; *Supplementary file 7* and *Supplementary file 8*. All dsDNA HDR templates were synthesized from IDT (0.25–10 µg). Unless otherwise noted, dsDNA HDR templates contained chemical modification that prevent unwanted integration events (e.g. non-homologous integration resulting in homology arm duplication or concatemerization) and boost HDR efficiency. These chemically modified templates can be purchased from IDT and are referred to as Alt-R HDR Donor Blocks. In this study, we tested knock-in efficiency comparing dsDNA HDR templates without chemical modification (i.e. IDT's standard gBlocks) to dsDNA HDR templates with two different proprietary chemical modifications. Of the two chemical modifications tested, only one is commercially available (the one in Alt-R HDR Donor Blocks) while the other early version of the chemical modification is not. In our experience, the Alt-R HDR Donor Blocks had the highest knock-in efficiency, and we used Alt-R HDR Donor Blocks for the majority of the HDR templates used in this study (*Figure 1—figure supplement 2A*). While Alt-R HDR Donor Blocks (combined with IDT's Alt-R HDR enhancer Version 2 [V2]) had the highest knock-in efficiency, all conditions tested worked with reasonably good efficiency (*Figure 1—figure supplement 2A*). Alt-R HDR Donor Blocks and gBlocks were resuspended in nuclease-free duplex buffer (IDT) to a concentration of 150 ng/µl and stored at –20 °C (*Supplementary file 1*).

## Preparation and microinjection of CRISPR/Cas9 reagents into African turquoise killifish embryos

We have included a detailed step-by-step protocol with all reagents, category numbers, and recipes in *Supplementary file 1*. Briefly, to prepare the gRNA complex the following were mixed: 1 µl 100 µM tracrRNA, 1 µl 100 µM crRNA, and 31.3 µl nuclease-free duplex buffer (IDT, 11-01-03-01) and annealed by incubation at 95 °C for 5 min. To form the ribonucleoprotein (RNP) complex, the following were mixed: 10 µl gRNA complex, 0.5 µl 10 µg/µl rCas9 protein (IDT, 1081059), and 5.5 µl 1x phosphate-buffered saline (1x PBS; Corning, 21–040-CV). This mixture was then incubated at 37 °C for 10 min. To prepare the injection mixture with the HDR template, the following were mixed: 8 µl RNP complex, 1 µl 150 ng/µl dsDNA HDR template (i.e. Alt-R HDR Donor Block), and 1 µl 10 µM small molecule HDR enhancer (IDT's Alt-R HDR enhancer Version 2 [V2]; IDT, 10007921). Finally, 0.33 µl of 8% phenol red was added to the injection mixture for visualization. The mixture was used immediately (within 1 hr of production) and kept on ice. Preassembled Cas9 RNP complex and synthetic dsDNA HDR templates were injected into the single cell of one-cell stage killifish embryos in accordance with microinjection procedures described in *Harel et al., 2015*. For each target locus and HDR template, between 32–249 embryos were injected per independent injection replicate (*Supplementary file 4*). Assuming similar embryo survival rates to what we observe in this study, we recommend injecting ~100 embryos

to achieve multiple positive injected animals that survive past development and can serve as F0 founders. Surviving injected embryos were maintained in embryo solution at 26–27 °C for 2–3 weeks. Embryos were then transferred to moist autoclaved coconut fiber (Eco Earth Coconut Fiber, EE-8) lightly packed in petri dishes where they were incubated for another 2–3 weeks at 26–27 °C after which they were hatched [as described in African turquoise killifish care and husbandry (*Bedbrook et al., 2023*; *Nath et al., 2023*)].

## Assessment of genome editing

Visual screening: Visual fluorescence screening of 14- to 21-day-old F0 embryos on a Fluorescent Stereo Microscope (Leica M165FC; *Figure 1B*) was conducted to verify successful knock-in of cDNA encoding fluorescent proteins. Twenty-one-day old embryos were dried on coconut fiber for 7 days prior to imaging.

Genotyping: PCR amplification of genomic DNA from fish tail clips or whole embryos was used to verify successful knock-in events. For this, we followed protocol described in *Hu et al., 2020*. Briefly, for tail clips, caudal fin clips were taken from larval (1 day post hatching) to 3-week-old fish. Tissue was digested in 30 µl DirectPCR Lysis Reagent (Mouse Tail) (Viagen, 102 T) with 40 µg/ml Proteinase K (Invitrogen, 25530049) at 55 °C for 2 hr followed by 100 °C heat inactivation for 10 min. This solution was used as template for PCR amplification with the following PCR reaction mixture (20 µl): 3 µl crude tissue lysis, 1 µl 10 µM primers (IDT), 10 µl 2x GoTaq Master Mixes (Promega, M7123), and 6 µl water. The PCR was run for 30–42 cycles. We used primer sets that enabled detection of genome editing based on amplification product size by gel electrophoresis (*Figure 1D*; *Figure 2C*; *Figure 2—figure supplement 1*; *Figure 3C*; *Figure 4A*; *Figure 4—figure supplement 1*). The primer sequences used to verify successful editing by genotyping are provided in *Supplementary file 3*.

Sequencing: To verify the sequence of successfully edited genomes and for assessment of potential off-target editing, PCR amplification of the genomic DNA was also sent for sequencing (Molecular Cloning Laboratories, MCLAB). The sequencing primer sequences used to verify successful editing are provided in *Supplementary file 3*. We have currently only tested for potential off-target editing for the two *ELAVL3* lines [*ELAVL3-T2A-Venus* (*Figure 2—figure supplement 2*) and *ELAVL3-T2A-Venus-P2A-H2B-oScarlet* (*Figure 3—figure supplement 1*)].

## Reverse transcription followed by quantitative PCR (RT-qPCR)

For these experiments, we focused on the *ELAVL3-T2A-Venus* line and the *NPY-T2A-Venus* line. To assess the impact of knock-in on the endogenous expression level of the targeted genes (e.g. *ELAVL3*, *NPY*), cohorts of F3 wildtype, heterozygous, and homozygous siblings were generated by crossing F2 heterozygous animals for each line. After genotyping, 4–6 animals per group were selected. For the *ELAVL3-T2A-Venus*-cross cohort, only females were selected to avoid any variability due to sex. Due to limits in the number of available animals for the *NPY-T2A-Venus*-cross cohort, both males and females were used but were balanced across genotypes. Select animals (4–6 animals per group) were aged to one month at which point their brains were harvested and snap-frozen in liquid nitrogen (stored at –80 °C until RNA isolation). RNA was isolated from brain tissue following protocol described in *McKay et al., 2022*. Briefly, brain tissue was transferred to 1.2 ml Collection Microtubes (QIAGEN, 19560). Autoclaved metal beads (QIAGEN, 69997) and 700 µl of QIAzol (QIAGEN, 79306) were added to each tube followed by tissue homogenization on a TissueLyserII machine (QIAGEN, 85300). Lysates was transferred to 1.5 ml tubes and 140 µl chloroform (Fisher Scientific, C298-500) was added followed by vortexing and incubation at room temperature for 2–3 min. Lysates were then centrifuged at 12,000×*g* at 4 °C for 15 min. The aqueous phase was mixed with 350 µl ethanol (200 Proof, Gold Shield Distributors, 412804) and then transferred to RNeasy columns from the RNeasy RNA Purification Kit (QIAGEN, 74106). Total brain RNA was then isolated according to the RNeasy RNA Purification Kit protocol.

For each sample, 1 µg of RNA was reverse-transcribed using the High Capacity cDNA Reverse Transcription Kit (Applied Biosystems, 4368814) according to the manufacturer's instructions. cDNA was diluted 1:10 and used for qPCR on a ABI QuantStudio 12 K Flex RT-PCR System using SYBR Green PCR Master Mix (Applied Biosystems, 4309155) according to the manufacturer's instructions. Beta actin (*ACTB*), glyceraldeyde-3-phosphate dehydrogenase (*GAPDH*), and elongation factor 1-alpha (*ELFA*) were used as housekeeping genes for normalization (using the geometric mean of all three

housekeeping genes). Primer pairs for *NPY*, *ELAVL3* and all housekeeping genes were designed such that they are separated by at least one intron. Given the extension time for the qPCR, this should eliminate amplification from any potential contaminating genomic DNA. For all primer sets used, we observed a single peak in the melt curve indicating a single amplification product. Primer sets used show no amplification in the no RT control. The amplification efficiency of all qPCR primer sets used was within the acceptable range of 90–110%. All primer sequences are listed in *Supplementary file 3*.

## Tissue histology

For brain sectioning, extracted whole brain samples from 1- to 4-month-old animals were fixed overnight in 4% paraformaldehyde in PBS (Santa Cruz Biotechnology, SC281692) at 4 °C and then washed for 12 hr in 1x PBS (Corning, 21–040-CV) at 4 °C with three washes. For sectioning larval stage killifish (1 day post hatching), whole animals were fixed overnight in 4% paraformaldehyde in PBS (Santa Cruz Biotechnology, SC281692) at 4 °C and then washed for 12 hr in 1x PBS (Corning, 21–040-CV) at 4 °C with three washes.

Fixed samples (either whole brains or whole larval fish) were dehydrated in 30% sucrose (Sigma-Aldrich, S3929) in 1x PBS at 4 °C overnight or until tissue sunk. Tissue was then embedded in Tissue-Plus OCT (Fisher Scientific, 23-730-571) within plastic embedding molds. Tissue was then frozen at –80 °C for at least 2 hr and sectioned (50–100 µm sections either sagittal or coronal) on a cryostat (Leica CM3050 S) and mounted on glass slides (Fisher Scientific, 12-550-15) and stored at –20 °C.

For immunofluorescence, slides were washed once in 1x PBS at room temperature to remove residual OCT. Slides were dehydrated and permeabilized in pre-chilled 100% methanol (Sigma-Aldrich, HPLC grade) with 1% Triton X-100 (Fisher Scientific, BP151) at –20 °C for 15 min, followed by washing in 1x PBS at room temperature. Slides were blocked with 5% Normal Donkey Serum (NDS; ImmunoReagents Inc, SP-072-VX10) and 1% Bovine Serum Albumin (BSA; Sigma, A7979) in 1x PBS ('blocking buffer') for 30 min at room temperature. Slides were washed in 1x PBS with 0.1% Tween-20 (PBST) three times for 10 min each, followed by washing in PBS. Slides were incubated in primary antibody (rabbit GFP Polyclonal Antibody, ThermoFisher, A-6455) at a 1:250 dilution in blocking buffer overnight at 4 °C followed by washing in PBST three times for 30 min each and then washing in 1x PBS. Slides were incubated in secondary antibody (donkey anti-rabbit IgG, ThermoFisher, A-31573) at a 1:500 dilution in blocking buffer for 2 hr at room temperature followed by washing in PBST three times for 30 min each and then washing in 1x PBS. Slices were mounted in either ProLong Gold Antifade Mountant (ThermoFisher, P36930) or ProLong Gold Antifade Mountant with DAPI (ThermoFisher, P36931) for imaging.

## *In situ* hybridization by hybridization chain reaction (HCR)

For *in situ* hybridization, we used *in situ* hybridization by hybridization chain reaction (HCR) following a protocol described in *Lovett-Barron et al., 2017*. First, hybridization probes were designed according to the split initiator approach of third generation *in situ* hybridization chain reaction (*Choi et al., 2018*), which enables automatic background suppression. Twenty-two-nucleotide long DNA antisense oligonucleotide split probes were designed for *Venus*, *ELAVL3*, *CRYAA*, *NPY*, and *HCRT* based on the killifish mRNA sequence (*Supplementary file 5*) and synthesized by IDT (200 µM in RNAse-free $H_2O$). Dye-conjugated hairpins (B1-647, B3-488 and B5-546) were purchased from Molecular Instruments. Slides were washed in 1x PBS at room temperature to remove residual OCT. Slides were dehydrated and permeabilized in pre-chilled 100% methanol (Sigma-Aldrich, HPLC grade) with 1% Triton X-100 (Fisher Scientific, BP151) at –20 °C for 15 min followed by washing three times in 2X saline sodium citrate (SSC) buffer with 0.1% Tween-20 (2x SSCT; made from 20x SSC, ThermoFisher, AM9763) at room temperature for 30 min each. Slides were equilibrated in hybridization buffer (2x SSCT, 10% (w/v) dextran sulfate [Sigma Aldrich, D6001], 10% (v/v) formamide [Thermo Fisher, AM9342]) for 30 min at 37 °C. Slides were then hybridized with split probes in hybridization buffer at a probe concentration of 4 nM overnight at 37 °C. Slides were then washed two times in 2x SSCT and 30% (v/v) formamide for 30 min at 37 °C. Slides were washed two times in 2x SSCT for 30 min each at room temperature. Slides were pre-amplified in amplification buffer (Molecular Instruments) for 10 min at room temperature. Dye-conjugated hairpins were prepared according to manufacturer's instructions. Briefly, they were heated to 95 °C for 1 min then snap-cooled to 4 °C. Amplification was performed by incubating slides in amplification buffer with prepared B1, B3 and/or B5 probes at concentrations of 120 nM overnight

in the dark at room temperature. Slides were washed 3 times with 2x SSCT for 30 min each. Slices were mounted in ProLong Gold Antifade Mountant for imaging.

### Whole-mount tissue clearing

For whole-mount tissue clearing (shown in *Figure 3E*), extracted whole brain samples from 1 to 4-month-old animals were fixed overnight in 4% paraformaldehyde in 1x PBS at 4 °C and then washed for 12 hr in 1x PBS at 4 °C with three washes. Fixed brain samples were crosslinked in a SHIELD hydrogel (*Park et al., 2018*) overnight in 1–2% SHIELD epoxide reagent (GE38; CVC Thermoset Specialties of Emerald Performance Materials) in 0.1 M Carbonate Buffer (pH 8.3) at 37 °C and then washed three times for 1 hr each in 1x PBS at 37 °C. Samples were cleared for 12–48 hr (depending on brain size) in 4% sodium dodecyl sulfate (SDS) at 37 °C until optically translucent and then washed three times for 1 hr intervals in 1x PBS with 0.1% Tween-20 (PBST) at 37 °C. For imaging, samples were then equilibrated in EasyIndex (RI = 1.52, LifeCanvas Technologies) and mounted.

### Imaging

All samples (unless otherwise noted) were imaged using an Olympus FV1200 confocal microscope system running Fluoview software, using a 10x0.6 Numerical Aperture water immersion Olympus objective. Images were collected at a 5 µm z-step resolution. Images in *Figure 2F–G*, *Figure 4C*, and *Figure 4E* were collected using an Olympus FV1200 confocal microscope system running Fluoview software, using a 20x1.0 Numerical Aperture water immersion Olympus objective (XLUMPLFLN Objective). Higher magnification images in *Figure 3D* were collected using a Zeiss LSM900 confocal microscope (Axio Observer) system running ZEN software (3.0, blue), using a 40x1.4 Numerical Aperture oil immersion Zeiss objective (Plan-Apochromat). Images were collected at a 4.5 µm z-step resolution. Single photon excitation was used at the indicated wavelengths. Entire samples were obtained by mosaic tiling during imaging, reconstructed using Fluoview software, and viewed and analyzed in Fiji and Aivia software.

## Acknowledgements

We thank Drs. Felix Boos, Jing Chen, Tyson Ruetz, Lucy Xu and John Bedbrook, as well as Charu Ramakrishnan, and all members of the Brunet lab and Deisseroth lab for their input on the project and providing feedback on the manuscript. We thank Dr. Lucy Xu for advice and guidance on RT-qPCR experiments and Dr. Ritchie Chen for advice and guidance on *in situ* hybridization methods. We thank IDT for generously providing reagents for testing. We thank Rogelio Barajas, Rishad Khondker, Jacob Chung, and Natalie Schmahl for killifish husbandry support. We thank Rogelio Barajas for managing the killifish room and help with development and maintenance of lines. This work was supported by R01AG063418 (AB and KD), the Glenn Foundation for Medical Research (AB), the Simons Foundation (AB), Chan Zuckerberg Biohub – San Francisco (AB), the Knight-Hennessy Scholars Graduate Fellowship (RN), the Helen Hay Whitney Postdoctoral Fellowship (CNB), the Wu Tsai Neurosciences Institute Interdisciplinary Scholar Award (CNB), T32 AG000266 (CNB), T32 AG0047126 (RDN), the Iqbal Farrukh & Asad Jamal Center for Cognitive Health in Aging (RDN), and a Brain Resilience Scholar Award from the Knight Initiative for Brain Resilience at the Wu Tsai Neurosciences Institute (RDN).

## Additional information

### Funding

| Funder | Grant reference number | Author |
| --- | --- | --- |
| National Institutes of Health | R01AG063418 | Karl Deisseroth Anne Brunet |
| Glenn Foundation for Medical Research | | Anne Brunet |
| Simons Foundation | | Anne Brunet |

| Funder | Grant reference number | Author |
|---|---|---|
| Chan Zuckerberg Biohub - San Francisco | | Anne Brunet |
| Knight-Hennessy Scholars Graduate Fellowship | | Rahul Nagvekar |
| Helen Hay Whitney Foundation | | Claire N Bedbrook |
| Wu Tsai Neurosciences Institute Interdisciplinary Scholar Award | | Claire N Bedbrook |
| National Institutes of Health | T32 AG000266 | Claire N Bedbrook |
| National Institutes of Health | T32 AG0047126 | Ravi D Nath |
| Iqbal Farrukh & Asad Jamal Center for Cognitive Health in Aging | | Ravi D Nath |
| Knight Initiative for Brain Resilience Scholar Award | | Ravi D Nath |

The funders had no role in study design, data collection and interpretation, or the decision to submit the work for publication.

## Author contributions

Claire N Bedbrook, Ravi D Nath, Conceptualization, Resources, Data curation, Formal analysis, Supervision, Funding acquisition, Validation, Investigation, Visualization, Methodology, Writing – original draft, Writing – review and editing; Rahul Nagvekar, Resources, Data curation, Formal analysis, Funding acquisition, Validation, Investigation, Visualization, Methodology, Writing – review and editing; Karl Deisseroth, Resources, Supervision, Funding acquisition, Writing – original draft, Project administration, Writing – review and editing; Anne Brunet, Conceptualization, Resources, Supervision, Funding acquisition, Writing – original draft, Project administration, Writing – review and editing

## Author ORCIDs

Claire N Bedbrook ⓘ http://orcid.org/0000-0003-3973-598X
Ravi D Nath ⓘ http://orcid.org/0000-0003-0905-2707
Rahul Nagvekar ⓘ http://orcid.org/0000-0001-8800-7503
Anne Brunet ⓘ http://orcid.org/0000-0002-4608-6845

## Ethics

This study was performed in strict accordance with the recommendations in the Guide for the Care and Use of Laboratory Animals of the National Institutes of Health. All animals were raised in accordance with protocols approved by the Stanford Administrative Panel on Laboratory Animal Care (protocol #APLAC-13645).

## Decision letter and Author response

Decision letter https://doi.org/10.7554/eLife.80639.sa1
Author response https://doi.org/10.7554/eLife.80639.sa2

# Additional files

## Supplementary files

- MDAR checklist
- Supplementary file 1. Step-by-step protocol for CRISPR/Cas9-mediated knock-in for killifish.
- Supplementary file 2. Annotated *ELAVL3-T2A-Venus* HDR sequence.
- Supplementary file 3. Sequences of primers, guide RNAs, and HDR repair templates.
- Supplementary file 4. Number of injected embryos, lethality, and knock-in efficiency for several loci

and different HDR donor templates with or without HDR enhancer.
- Supplementary file 5. Probe sets for *in situ* hybridization chain reaction.
- Supplementary file 6. Annotated *ELAVL3-T2A-Venus-P2A-H2B-oScarlet* HDR sequence.
- Supplementary file 7. Annotated *NPY-T2A-Venus* HDR sequence.
- Supplementary file 8. Annotated *HCRT-T2A-Venus* HDR sequence.

## Data availability

All data generated or analyzed during this study are included in the manuscript and supporting files are provided.

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
