## [Editor Report]

This paper describes a rapid and easy to implement CRISPR/Cas9-mediated knock-in approach to precisely insert large transgenes in the African turquoise killifish. The methodologies performed are rigorous and the conclusions reached are well supported by the data. The established method is instrumental for many researchers working with unusual model species, and, in particular will expand the killifish community toolbox. It will revolutionize the field and bring the killifish, an emerging animal model in aging biology and disease modeling in vertebrates, into the spotlight even more.

---

## [Decision Letter]

**Decision letter after peer review:**

Thank you for submitting your article "Rapid and precise genome engineering in a naturally short-lived vertebrate" for consideration by *eLife*. Your article has been reviewed by 3 peer reviewers, including Lieve Moons as Reviewing Editor and Reviewer #1, and the evaluation has been overseen by Didier Stainier as the Senior Editor. The following individual involved in the review of your submission has agreed to reveal their identity: Phil Abitua, PhD (Reviewer #3).

Essential revisions:

Although the reported data are of major interest and relevance to the field, they are as yet not sufficiently supporting all conclusions made. The manuscript would benefit from the inclusion of a more detailed material and methods section, more rigorous validation of the KI lines that were generated, an elaboration on (morphometrical) expression data, and a more critical discussion.

Specifically

1. More detailed information is needed in the material and methods section, such as more information on the chemical modification of the repair template (reveal the modification or use the appropriate IDT reference number), and a complete description of the used solution recipes. Although full disclosure of the methodology might not be possible, as it is described now, it does not allow replication.

2. The authors should perform extra co-stainings that link endogenous expression to the reporter fluorescence (co-localization of GFP with HCR for the targeted genes), preferably for all lines, and determine whether the KI affects the endogenous expression level of the targeted genes.

3. The authors should add figures (e.g., in supplement) that reveal the expression of all studied genes, at minimum in larval fish.

4. The authors should insert a (supplemental) table that clearly outlines the initial number of eggs injected, the number of surviving embryos, the number of positive embryos, etc. for all lines displayed in the paper. They are encouraged to include for some of the lines a larger number of animals for the quantifications of the targeting efficiency and embryo survival. As these are now based on rather limited numbers for some lines and the analysis of germline TM has only been shown for 1 line, the claims made seem too strong, and should be rephrased/tempered where needed. As an example: "We observe efficient CRISPR/Cas9-mediated knock-in of large inserts (>40% efficiency) with germline transmission rates over 65%" is an overinterpretation if it is just based on ELAVL.

5. The authors are encouraged to strengthen their discussion by including a more detailed comparison of their method to other CRISPR-based KI methods in more typical model species, by putting the work in context with the GeneWeld paper from Wierson et al. 2020 and the method described by Seleit et al. 2022, and by contrasting the advantages and disadvantages of the used methodology.

*Reviewer #1 (Recommendations for the authors):*

The data are of major interest and relevance to the scientific community. Although the manuscript does not report on thrilling innovative findings, it provides an efficient and amendable knock-in approach that paves the way for future studies in the killifish animal model system.

The following issues could be addressed:

– Although all components of the method can be ordered at IDT, there is a big lack in the transparency of the kind of modification that is used and the composition of the HDR-enhanced buffer. As the composition of the HDR-enhanced buffer will never be disclosed by IDT, it might be useful to share a mode of action on how this blocks non-homologous end joining (NHEJ).

– The conclusion that modification #3 in combination with HDC enhancer results in the most optimal efficiency seems rather premature. To make this statement, preferably more technical repeats should be conducted (lines 88 – 93). A similar comment can be made for the survival rate where only two technical replicates were used. Moreover, the number of embryos evaluated per experiment differs from 10 to 105. This gap is too large and therefore it should at least be mentioned where or for which constructs only 10 embryos were scored.

– One other interesting experiment would be to see if the efficiency of germline transmission is promotor (region) dependent. Here, a general conclusion is made that germline transmission using the described technique is high, but it might be lower in heterochromatin regions. This point should at minimum be discussed.

– A "successful" attempt was made to create homozygous F1s (using 1 pair of F0 fish). However, how the F0s for this homozygous crossing were chosen is not disclosed. Moreover, silent mutations might be selected without backcrossing, thereby downscaling the use of F1 homozygotes. Although briefly acknowledged by the authors later in the manuscript, a more elaborate discussion of the benefits and risks should be provided.

– Correct expression of transgenes in the larval stage is only shown for ELAVL3-T2A-Venus, it would be nice to also see it for the other constructs, also as this should enable direct comparison to zebrafish work. Preferably, although not really needed in this paper, also the expression in the adult fish should be reported/shown, also as to reveal the activity of the promotors in the adult fish.

– During the insertion of a long (1.8kb) construct the nuclear localization signal (NLS) from histone 2B is used to target oScarlet to the nucleus. Authors could disclose whether the sequence is derived from killifish or zebrafish/medaka? In the final manuscript, the repair templates should be annotated as well, instead of just providing the sequence (homology arms, self-cleaving peptides, fluorophores, localization sequences).

– To evaluate if the insertion of the longer construct is as efficient as the shorter construct more technical replicates seem needed. Moreover, survival rates of injected embryos with the long construct next to non-injected embryos should preferably be disclosed within the manuscript itself to evaluate construct toxicity.

– The authors indicate that expression of ELAVL3-T2A-Venus-P2A-H2B-oScarlet is likely limited to neurons (line 147). This can easily be confirmed by doing staining for HuC/D on these transgenic fish, which is required if authors want to claim that neuronal-specific labeling is possible using this technique (line 154). Also, based on the DAPI staining, it is clear that not all cells (neurons) are labeled. It would thus be interesting to know the percentage of neurons labeled and compare it to Elav labeling in adult zebrafish?

– Would it be possible to also show Elav expression in the adult eye/retina and/or spinal cord?

– Throughout the manuscript, the effect of insertion on endogenous protein expression levels is not evaluated. Here, the authors could perform qPCR or Western blotting for the endogenous proteins (e.g. Npy, Hcrt, Elavl3, ….) to compare expression levels between WT and transgenic fish.

– In lines 46 and 203, the authors say that this method can "allow the establishment of versatile strategies to probe complex phenotypes, including development, 'suspended animation', regeneration, aging, and age-related diseases." Authors should maybe briefly specify what they mean by "suspended animation". Another word that should be explained is "landing pad" (line 219).

– gRNA is composed of tracrRNA and crRNA, however, many experts in the field are using sgRNA's. What is the advantage of using tracrRNA and crRNA over sgRNA's? Have sgRNA's been evaluated as well? A short discussion on this topic would be appreciated.

*Reviewer #2 (Recommendations for the authors):*

RE "a more rigorous validation of the KI lines that were generated".

It is important to measure whether the edit, which is done at the 3' end, disrupts gene regulation and potentially endogenous gene expression levels. Is the endogenous expression changed by the insertion of the fluorescent tag? Often HDR is accompanied by additional small edits around the targeted site. Did the authors check for these small edits in the 3'UTR or the wider non-coding region?

Regarding ectopic expression, the authors mention that "In all embryos screened, we did not observe Venus expression in a tissue that was not specifically targeted." The drawback of screening killifish eggs during development is that only larger misexpression can be detected because of the chorion of the egg that is not fully transparent. Did the authors perform a more extensive screen of the faithfulness of expression in F1 or F2 (homozygous) animals? In Figure 2, please add images for other strains (CRYAA and ACTB) to demonstrate the specificity of expression.

Figure 1D shows that wt and ki bands are detected in the F0 animal, which could be a sign of heterozygosity or mosaicism. Can the authors distinguish and potentially estimate the percentage of mosaicism?

To prove that the targeted promotor faithfully reflects gene expression, a co-staining of the reporter with endogenous gene expression should be done. For the moment, the HCRs shown in Figure4-suppl2 do not have a lot of added value and should be performed in the KI lines.

Regarding "rather limited numbers of animals"

This reviewer found it difficult to get a full view of the experimental flow with regard to animal numbers, as information related to efficiency and lethality was spread out in different places. I would recommend making a merged supplemental table displaying the initial number of eggs injected, the number of surviving embryos, the number of positive embryos, etc. for all lines displayed in the paper. This will give a better view of the experimental setup and the efficiency of the method. The number of animals should be larger (minimally 50 surviving) to strengthen the efficiency claim. Germline TM was only shown for 1 line. Did the other lines go germline as well? Does the KI have an effect on fertility?

Lethality is high; at what stage did the authors lose the embryos?

Regarding " Figures of the manuscript contain information that is not discussed in the main text"

Figure 1D: bands seem to suggest that some F0 are homozygous? What is the percentage of ho in F0? 85% of the fish are homozygous after crossing 2 different F0: these percentages suggest that the F0 might already be edited on both alleles. Editing on both alleles might also have created different additional small edits that, when combined, have a potentially larger effect on gene expression. Have the authors seen such effects in the combined F1 generation?

Figure 2E: green signal apparent in WT in the somites? This is not apparent in all pictures, but I think the authors might have an explanation for this green signal.

Figure3D/E: ELAVL3 is expected to drive expression in all postmitotic neurons. Expression is expected to be similar in all postmitotic neurons, however distinct non-expressing stripes can be observed. This seems to indicate a mosaic situation with some (clonally related?) cells not expressing ELAVL3, but it is unclear how this can appear in an F1 animal (D). Do the authors have an explanation for this phenomenon?

Figure 3E Expression in brain hemispheres is not identical is some regions; This is an F0 animal that could have mosaicism; can this be repeated in F1 or F2? Why the different ages? Can the age of the animal have an effect on the expression levels of ELAVL3?

Regarding: "Full disclosure of methods"

The authors mention chemical modifications of the templates #1 and #3 but no more information was given. The IDT website does not mention these modification options, so it will be advisable to reveal the modification or use the appropriate IDT reference number. Used solution recipes should be complete.

*Reviewer #3 (Recommendations for the authors):*

I recommend this manuscript for publication but have a few questions/comments after reading the paper.

1) I think the paper would benefit from putting the work in context with the GeneWeld paper from Wierson et al. 2020 and the method described by Seleit et al. 2022. Contrasting the advantages and disadvantages of this work with theirs would be helpful.

2) Can the authors comment on why they used 150-200 bp homology arms?

3) I am left wondering how much of the efficiency reported in this paper is due to the locus being targeted. The presence of a PAM sequence near the endogenous stop codon is a limitation of the technique. Are their ways of circumventing this or alternative strategies? This would be a good discussion point.

4) On a related note, are the distances between the Cas9 cut sites and target insertion sites reported in the paper?

5) It is difficult to interpret supplemental figure 1A since two chemical modifications are used. It would be nice if the panel included modification #3 without HDR enhancer.

6) 1XPBS should have a space. 1X PBS?

---

## [Author Response]

Essential revisions:1. More detailed information is needed in the material and methods section, such as more information on the chemical modification of the repair template (reveal the modification or use the appropriate IDT reference number), and a complete description of the used solution recipes. Although full disclosure of the methodology might not be possible, as it is described now, it does not allow replication.

We thank the Reviewers for these helpful comments. As suggested, we have now included a step-by-step protocol with all reagents and recipes including the IDT reference number (when available) and product names (new Supplementary File 1). We have also included additional details in the material and methods section. Furthermore, to avoid confusion about reagent information, we no longer refer to the chemical modification on the HDR template as “modification #3”, as this is not the term used for the commercially available product. We now refer to it throughout the revised manuscript as “Alt-R HDR Donor Block” from IDT (while this product does not have a specific catalog number because they are customized, we provide the IDT URL for ordering this type of HDR template from IDT). We now also provide the catalog number for the HDR enhancer from IDT. We believe that our revised material and method section and the inclusion of a step-by-step protocol with all the commercially available reagents greatly improves our manuscript. We thank the Reviewers for these suggestions, as we agree that it will help replication (and indeed this method has now been successfully used by all 4 members of the Brunet lab who have tried it and 1 member from another lab).

2. The authors should perform extra co-stainings that link endogenous expression to the reporter fluorescence (co-localization of GFP with HCR for the targeted genes), preferably for all lines, and determine whether the KI affects the endogenous expression level of the targeted genes.

We agree with the Reviewers that these are important experiments. We have now performed staining to test co-expression of fluorescent reporter transcript (*Venus*) and the transcript of the target loci (e.g., *ELAVL3, CRYAA*, *NPY*, and *HCRT*) via *in situ* hybridization using hybridization chain reaction (HCR) (new Figure 2F, Figure 4C, Figure 4E, and Figure 1— figure supplement 1). We observe that the *Venus* reporter recapitulates the expression pattern of the endogenous target transcript in the knock-in lines *ELAVL3-T2A-Venus, NPYT2A-Venus*, and *HCRT-T2A-Venus* (in heterozygous animals for all lines). The *Venus* reporter also recapitulates the expression pattern of the target transcript in Venus positive F0 animals targeting the CRYAA locus. We also find that *Venus* expression in each transgenic is similar to the expression of the target gene in the context of wildtype individuals for *ELAVL3* and *CRYAA* (new Figure 2G and Figure 1—figure supplement 1), which we had previously verified for *NPY* and *HCRT* (Figure 4—figure supplement 2). Together, these results indicate that the Venus reporter recapitulates the expression of the endogenous genes, and they also suggest that the endogenous allele is not grossly disrupted.

To examine in more depth the impact of the transgene on the endogenous allele, we also used a more quantitative RT-qPCR approach to compare *Venus* transcript expression and endogenous target expression for two stable lines generated in the paper (*ELAVL3-T2AVenus* and *NPY-T2A-Venus*) (new Figure 2—figure supplement 3 and Figure 4—figure supplement 3). We compared expression in wildtype, heterozygous, and homozygous siblings (generated by crossing heterozygous animals). We find that, as expected, Venus expression is only observed in heterozygous and homozygous animals in both lines (with increased expression in homozygous). Expression of endogenous *ELAVL3* was not affected by the presence of the transgene in the *ELAVL3-T2A-Venus*, whether homozygous or heterozygous (although there was a small non-significant decrease in homozygous) (new Figure 2—figure supplement 3). Expression of endogenous *NPY* was not affected by the presence of the transgene in the *NPY-T2A-Venus* in heterozygous individuals, but it was slightly and significantly decreased in homozygous individuals (new Figure 4—figure supplement 3). This indicates that the presence of the transgene can decrease the expression at homozygous level. We have included these new experiments in the revised manuscript, and we also include recommendations to generate and use such stable lines as heterozygous.

3. The authors should add figures (e.g., in supplement) that reveal the expression of all studied genes, at minimum in larval fish.

We thank the Reviewers for this suggestion. We have now done additional HCR experiments for key endogenous target genes (*ELAVL3*, *CRYAA*, *NPY*, and *HCRT*) in both wildtype animals (new Figure 2G and Figure 1—figure supplement 1) and knock-in lines (new Figure 2F, Figure 4C, Figure 4E, and Figure 1—figure supplement 1). Unfortunately, we were not able to do HCR experiments for *ACTB2* in the knock-in line, because the knock-in targeting *ACTB2* was lethal at the F0 stage [consistent with what has been observed in Medaka (Gutierrez-Triana et al., 2018)]. We have now indicated this clearly in the revised manuscript. We have included the new staining in Figure 2F–G, Figure 4C, Figure 4E, and Figure 1— figure supplement 1 in the revised manuscript.

4. The authors should insert a (supplemental) table that clearly outlines the initial number of eggs injected, the number of surviving embryos, the number of positive embryos, etc. for all lines displayed in the paper. They are encouraged to include for some of the lines a larger number of animals for the quantifications of the targeting efficiency and embryo survival. As these are now based on rather limited numbers for some lines and the analysis of germline TM has only been shown for 1 line, the claims made seem too strong, and should be rephrased/tempered where needed. As an example: "We observe efficient CRISPR/Cas9-mediated knock-in of large inserts (>40% efficiency) with germline transmission rates over 65%" is an overinterpretation if it is just based on ELAVL.

We thank the Reviewers for this helpful feedback. We have now done the following experiments to address the Reviewers’ suggestions:

1. We have now performed additional rounds of injections for all injection conditions quantified in the paper (with the exception of *ACTB2-T2A-Venus* knock-in as they did not survive hatching in F0s). For each injection condition, we preformed between 3–6 independent replicate rounds of injection. In total, we performed >30 separate rounds of injections and injected >3000 embryos. With the additional replicates, we have a total of between 43–156 developed embryos per injection condition. The larger number of replicates enabled us to perform statistical comparison of the knock-in efficiency of “chemically-modified dsDNA HDR template with small molecule HDR enhancer” versus “unmodified HDR template without enhancer” for the *ELAVL3* locus (revised Figure 1C; Mann-Whitney test). These results indicate that “chemicallymodified dsDNA HDR template with small molecule HDR enhancer” is significantly more efficient compared to “unmodified HDR template without enhancer” at least for the *ELAVL3* locus (and we have decided to use this condition for all other knock-in experiments).

2. We expanded Supplementary File 4 to include all injections quantified in this paper to indicate (in a single table) the number of injected embryos, number of surviving embryos, number of developed vs diapausing embryos, and the number of Venus positive embryos. Based on these values, we have added guidelines in the detailed method section (new Supplementary File 1) and in the revised manuscript for the target number of embryos that should be injected for best practices.

3. We further characterize the lethality following knock-in injections by tracking embryo death for up to two weeks post fertilization comparing embryos injected with knock-in reagents to non-injected embryos from the same clutch (Figure1—figure supplement 2B,C). In our hands, killifish embryo survival is highly variable and clutch dependent, and we observe high lethality of fertilized embryos in both injected and not-injected (wildtype) conditions. Embryo survival, in general, is a challenge we have been facing in the killifish (even for non-injected wildtype embryos) and to overcome this, we recommend injecting ~100 embryos for each knock-in line to ensure successful generation of a transgenic embryos that survives to development. We have indicated this in the revised manuscript.

5. The authors are encouraged to strengthen their discussion by including a more detailed comparison of their method to other CRISPR-based KI methods in more typical model species, by putting the work in context with the GeneWeld paper from Wierson et al. 2020 and the method described by Seleit et al. 2022, and by contrasting the advantages and disadvantages of the used methodology.

Thank you for this important suggestion. We agree that this is a valuable addition to the paper. We have expanded the discussion to include details of advantages and disadvantages of our presented method in the context of these key papers (and additional papers).

We thank the Reviewers for constructive feedback, and we believe that our revised manuscript has improved as a result of these changes.

Reviewer #1 (Recommendations for the authors):The data are of major interest and relevance to the scientific community. Although the manuscript does not report on thrilling innovative findings, it provides an efficient and amendable knock-in approach that paves the way for future studies in the killifish animal model system.The following issues could be addressed:– Although all components of the method can be ordered at IDT, there is a big lack in the transparency of the kind of modification that is used and the composition of the HDR-enhanced buffer. As the composition of the HDR-enhanced buffer will never be disclosed by IDT, it might be useful to share a mode of action on how this blocks non-homologous end joining (NHEJ).

The Reviewer’s input is well taken. Unfortunately, we do not have access to information on the specific mode of action of these IDT reagents beyond what was already in the manuscript. However, we now have included more detailed information on the IDT product numbers and product names in our step-by-step protocol (Supplementa File 1) and in the material and methods. We also note that while we do see a significant boost in efficiency of CRISPR/Cas9mediated knock-in using chemically modified HDR templates and IDT’s small molecule HDR enhancer, we still observe reasonably good knock-in efficiency without (Figure 1C). Thus, in the (hopefully unlikely) scenario where these reagents would no longer be commercially available from IDT, the knock-in protocol we have presented in this paper would still be efficient.

– The conclusion that modification #3 in combination with HDC enhancer results in the most optimal efficiency seems rather premature. To make this statement, preferably more technical repeats should be conducted (lines 88 – 93). A similar comment can be made for the survival rate where only two technical replicates were used. Moreover, the number of embryos evaluated per experiment differs from 10 to 105. This gap is too large and therefore it should at least be mentioned where or for which constructs only 10 embryos were scored.

The Reviewer’s feedback is well taken.

As suggested, we have now performed additional rounds of injections for all injection conditions quantified in the paper (with the exception of *ACTB2-T2A-Venus* knock-in as they did not survive hatching in F0s). For each injection condition, we performed between 3–6 independent replicate rounds of injection. In total, we conducted >30 separate rounds of injections and injected >3000 embryos. With the additional replicates, we have a total of between 43–156 developed embryos per injection condition. The larger number of replicates enabled us to perform statistical comparison of the knock-in efficiency of “chemically-modified dsDNA HDR template with small molecule HDR enhancer” versus “unmodified HDR template without enhancer” for the *ELAVL3* locus (revised Figure 1C; Mann-Whitney test). These results indicate that “chemically-modified dsDNA HDR template with small molecule HDR enhancer” is significantly more efficient compared to “unmodified HDR template without enhancer” at least for the *ELAVL3* locus (and we have decided to use this condition for all other knock-in experiments).

We expanded Supplementary File 4 to include all injections quantified in this paper to indicate (in a single table) the number of injected embryos, number of surviving embryos, number of developed vs diapausing embryos, and the number of Venus positive embryos. Based on these values, we have added guidelines in the detailed method section (Supplementary File 1) and in the revised manuscript for the target number of embryos that should be injected for best practices.

We have also characterized the lethality following knock-in injections by tracking embryo death for up to two weeks post fertilization comparing embryos injected with knock-in reagents to non-injected embryos from the same clutch (Figure1—figure supplement 2B,C). In our hands, killifish embryo survival is highly variable and clutch dependent, and we observe high lethality of fertilized embryos in both injected and not-injected (wildtype) conditions. Embryo survival, in general, is a challenge we have been facing in the killifish (even for noninjected wildtype embryos) and to overcome this, we recommend injecting ~100 embryos for each knock-in line to ensure successful generation of a transgenic embryos that survives to development. We have indicated this in the revised manuscript.

– One other interesting experiment would be to see if the efficiency of germline transmission is promotor (region) dependent. Here, a general conclusion is made that germline transmission using the described technique is high, but it might be lower in heterochromatin regions. This point should at minimum be discussed.

The reviewer brings up an important point. We have added a discussion of this point in the discussion of the paper:

“CRISPR/Cas9-mediated knock-in efficiency and germline transmission rates may vary dependent on the genomic locus being targeted (Hsu et al., 2013; Labun et al., 2019), which might be due to the chromatin accessibility of that locus”.

– A "successful" attempt was made to create homozygous F1s (using 1 pair of F0 fish). However, how the F0s for this homozygous crossing were chosen is not disclosed. Moreover, silent mutations might be selected without backcrossing, thereby downscaling the use of F1 homozygotes. Although briefly acknowledged by the authors later in the manuscript, a more elaborate discussion of the benefits and risks should be provided.

The Reviewer has another great point. We used one Venus-positive F0 male (F0 #1; Figure 2B) and one Venus-positive F0 female (F0 #2; Figure 2B), both of which produced a large fraction of Venus-positive progeny when crossed with wildtype fish (Figure 2B). We had multiple Venus-positive F0 males that produced many Venus-positive progeny. We randomly selected from those males. We have now better phrased this in the revised manuscript. We have also expanded on the discussion of the risks/benefits of this approach in the main text, emphasizing that this could lead to passing down of silent mutations and highlighting that the best practice would be to backcross F0 founders to wildtype (GRZ) animals.

“This accelerated inter-crossing approach could enable rapid testing of homozygous F1 lines if desired. However, we also note that directly generating homozygous F1 lines may also increase the risk of propagating silent mutations from F0 animals (e.g., off-target genome editing in F0s), which might in turn lead to phenotypes independent of the introduced transgene. We formally tested for potential off-target insertions/mutations upon CRISPR/Cas9-mediated knock-in. PCR amplification and Sanger sequencing of homozygous F1 *ELAVL3*-*T2A*-*Venus* animals at the three most likely off-target sites (predicted by CHOPCHOP (Labun et al., 2019)) showed no off-target editing at these sites (Figure 2—figure supplement 2). Nevertheless, to limit the risk of off-target editing, we recommend backcrossing founders to wildtype animals.”

– Correct expression of transgenes in the larval stage is only shown for ELAVL3-T2A-Venus, it would be nice to also see it for the other constructs, also as this should enable direct comparison to zebrafish work. Preferably, although not really needed in this paper, also the expression in the adult fish should be reported/shown, also as to reveal the activity of the promotors in the adult fish.

We thank the Reviewer for this suggestion. We have now performed HCR to test coexpression of the endogenous *CRYAA* transcript and the introduced transgene transcript (i.e., *Venus*) in both wildtype and *CRYAA-T2A-Venus* F0s in larval fish (new Figure 1—figure supplement 1). Additionally, we performed HCR to assess the target transcripts (e.g., *ELAVL3*, *NPY*, and *HCRT*) and the introduced transgene transcript (i.e., *Venus*) for stable lines of *ELAVL3*-*T2A*-*Venus, NPY*-*T2A*-*Venus, and HCRT*-*T2A*-*Venus* and in wildtype animals for the adult brain (new Figure 2F-G, Figure 4C and Figure 4E).

– During the insertion of a long (1.8kb) construct the nuclear localization signal (NLS) from histone 2B is used to target oScarlet to the nucleus. Authors could disclose whether the sequence is derived from killifish or zebrafish/medaka? In the final manuscript, the repair templates should be annotated as well, instead of just providing the sequence (homology arms, self-cleaving peptides, fluorophores, localization sequences).

The Reviewer’s point is well taken. We have now included details of the origin of the NLS sequence used in the main text of the paper:

“The oScarlet was also tagged with the nuclear localization signal (NLS) from human Histone 2B, a NLS commonly used for zebrafish transgenics (Freeman et al., 2014; Kanda et al., 1998).”

In addition, for each of the stable lines generated in this paper, we have included gb files with the repair templates with annotated homology arms, self-cleaving peptides, fluorescent proteins, and localization signals (Supplementary File 2 and 6–8).

– To evaluate if the insertion of the longer construct is as efficient as the shorter construct more technical replicates seem needed. Moreover, survival rates of injected embryos with the long construct next to non-injected embryos should preferably be disclosed within the manuscript itself to evaluate construct toxicity.

We thank the Reviewer for this suggestion. We have now performed additional replicates for both the long construct and short construct knock-in and observed consistent results (revised Figure 3B). We also included the lethality comparison between the long construct and short construct knock-in (new Figure 1—figure supplement 2B,C and Supplementary File 4). These results indicate that there is no increased toxicity with the long knock-in construct.

– The authors indicate that expression of ELAVL3-T2A-Venus-P2A-H2B-oScarlet is likely limited to neurons (line 147). This can easily be confirmed by doing staining for HuC/D on these transgenic fish, which is required if authors want to claim that neuronal-specific labeling is possible using this technique (line 154). Also, based on the DAPI staining, it is clear that not all cells (neurons) are labeled. It would thus be interesting to know the percentage of neurons labeled and compare it to Elav labeling in adult zebrafish?

This is another great suggestion. We performed HCR to test co-expression of *ELAVL3* (which encodes the neuronal protein HuC) and *Venus* in the stable *ELAVL3-T2A-Venus* transgenic animals and we observe a clear overlap. While *ELAVL3* is not a completely independent marker in this case (as it is both the targeted locus and the cell marker), we believe that these observations are consistent with positive cells being neurons.

The DAPI positive, *ELAVL3/Venus* negative cells are likely other brain cell types (e.g. astrocytes, oligodendrocytes). We agree that a comparison of the fraction of *ELAVL3* positive cells in the brain between killifish and zebrafish would be interesting, though we feel it would be outside of the scope of this paper. We have added a sentence to the legend of Figure 3 indicating the possible cell type for DAPI positive, *ELAVL3/Venus* negative cells.

– Would it be possible to also show Elav expression in the adult eye/retina and/or spinal cord?

We agree with the Reviewer this would be an interesting experiment. However, we have not done it yet, as we felt it was outside the scope of the present manuscript. In larval killifish, we do observe expression of the knock-in reporter in the spinal cord and retina (Figure 2E). Given that the adult retina and spinal cord are neuron-dense structures, we expect this expression pattern to persist in adult animals. We have now added a sentence in the text to reflect this.

– Throughout the manuscript, the effect of insertion on endogenous protein expression levels is not evaluated. Here, the authors could perform qPCR or Western blotting for the endogenous proteins (e.g. Npy, Hcrt, Elavl3, ….) to compare expression levels between WT and transgenic fish.

The Reviewer’s point is well taken. To examine in more depth the impact of the transgene on the endogenous allele, we have used an RT-qPCR approach to compare *Venus* transcript expression and endogenous target expression for the two stable lines generated in the paper (*ELAVL3-T2A-Venus* and *NPY-T2A-Venus*) (new Figure 2—figure supplement 3 and Figure 4—figure supplement 3). We compared expression in wildtype, heterozygous, and homozygous siblings (generated by crossing heterozygous animals). Expression of endogenous *ELAVL3* was not affected by the presence of the transgene in the *ELAVL3-T2AVenus*, whether homozygous or heterozygous (although there was a small non-significant decrease in homozygous individuals) (new Figure 2—figure supplement 3). Expression of endogenous *NPY* was not affected by the presence of the transgene in the *NPY-T2A-Venus* in heterozygous individuals, but it was slightly and significantly decreased in homozygous individuals (new Figure 4—figure supplement 3). This result indicates that the presence of the transgene can decrease the expression at homozygous level. We have included these new experiments in the revised manuscript and we also include recommendations to generate and use such stable lines as heterozygous rather than homozygous lines.

– In lines 46 and 203, the authors say that this method can "allow the establishment of versatile strategies to probe complex phenotypes, including development, 'suspended animation', regeneration, aging, and age-related diseases." Authors should maybe briefly specify what they mean by "suspended animation". Another word that should be explained is "landing pad" (line 219).

Thank you for bringing this to our attention. We have edited the text to now read:

“This genetic toolkit has enabled discoveries about the mechanisms of aging, regeneration, evolution, development, and embryonic diapause – a state of ‘suspended animation’.

“Additional variations, such as the use of ‘landing pads’ (i.e., genetic loci that support stable long-term expression of transgenes such as *ROSA26* in mice) (Soriano, 1999)”

– gRNA is composed of tracrRNA and crRNA, however, many experts in the field are using sgRNA's. What is the advantage of using tracrRNA and crRNA over sgRNA's? Have sgRNA's been evaluated as well? A short discussion on this topic would be appreciated.

Thank you for bringing this point up. We have included a discussion of this in the methods section of the manuscript:

“In this work, we use the two-part synthetic tracrRNA and crRNA purchased from IDT to generate a functional gRNA. This system is more cost-effective than purchasing the full length synthetic single guide RNA (sgRNA). Another cost-effective option is to perform in-house synthesis of sgRNA via in vitro transcription (e.g., using MEGAscript T7 Transcription Kit), though this requires more hands-on time. While sgRNAs have been used and validated in killifish for CRISPR/Cas9-mediated engineering (Harel et al., 2015; Harel et al., 2016), we did not evaluate sgRNAs for CRISPR/Cas9-mediated knock-in in this study.”

Reviewer #2 (Recommendations for the authors):RE "a more rigorous validation of the KI lines that were generated".It is important to measure whether the edit, which is done at the 3' end, disrupts gene regulation and potentially endogenous gene expression levels. Is the endogenous expression changed by the insertion of the fluorescent tag? Often HDR is accompanied by additional small edits around the targeted site. Did the authors check for these small edits in the 3'UTR or the wider non-coding region?

The Reviewer’s point is well taken. To examine the impact of the transgene on the endogenous allele, we used an RT-qPCR approach to compare *Venus* transcript expression and endogenous target expression for the two stable lines generated in the paper (*ELAVL3T2A-Venus* and *NPY-T2A-Venus*) (new Figure 2—figure supplement 3 and Figure 4—figure supplement 3). We compared expression in wildtype, heterozygous, and homozygous siblings (generated by crossing heterozygous animals). We find that, as expected, Venus expression is only observed in heterozygous and homozygous animals in both lines (with increased expression in homozygous individuals). Expression of endogenous *ELAVL3* was not affected by the presence of the transgene in the *ELAVL3-T2A-Venus*, whether homozygous or heterozygous (although there was a small non-significant decrease in homozygous individuals) (new Figure 2—figure supplement 3). Expression of endogenous *NPY* was not affected by the presence of the transgene in the *NPY-T2A-Venus* in heterozygous individuals, but it was slightly and significantly decreased in homozygous individuals (new Figure 4—figure supplement 3). This result indicates that the presence of the transgene can decrease the expression at homozygous level. We have included these new experiments in the revised manuscript and we also include recommendations to generate and use such stable lines as heterozygous rather than homozygous lines.

For all stable lines, we have sequenced genomic DNA external to the knock-in site (including at the 3’ end) and observed no mutations. For example, for the *ELAVL3* knock-in lines, we sequenced 500 bp 5’ and 500 bp 3’ of the knock-in site and observed no mutations. We have edited the text to clarify this point:

“PCR amplification and genotyping by Sanger sequencing of homozygous F1 animals confirmed that the *T2A*-*Venus* integration at the *ELAVL3* locus was as designed— single copy and in frame, with no observed mutations within 1 kb around the insertion site”

Regarding ectopic expression, the authors mention that "In all embryos screened, we did not observe Venus expression in a tissue that was not specifically targeted." The drawback of screening killifish eggs during development is that only larger misexpression can be detected because of the chorion of the egg that is not fully transparent. Did the authors perform a more extensive screen of the faithfulness of expression in F1 or F2 (homozygous) animals?

We thank the Reviewer for this important point. We have now performed staining to test coexpression of fluorescent reporter transcript (*Venus*) and the target transcripts (e.g., *ELAVL3, CRYAA*, *NPY*, and *HCRT*) via *in situ* hybridization using hybridization chain reaction (HCR) (new Figure 2F, Figure 4C, Figure 4E, and Figure 1—figure supplement 1). We observe that the *Venus* reporter recapitulates the expression pattern of the endogenous target transcript in the knock-in lines *ELAVL3-T2A-Venus, NPY-T2A-Venus*, and *HCRT-T2A-Venus* (in heterozygous animals for all lines). The *Venus* reporter also recapitulates the expression pattern of the target transcript in Venus positive F0 animals targeting the CRYAA locus. We also find that *Venus* expression in each transgenic is similar to the expression of the target gene in the context of wildtype individuals for *ELAVL3* and *CRYAA* (new Figure 2G and Figure 1—figure supplement 1), which we had previously verified for *NPY* and *HCRT* (Figure 4— figure supplement 2). Together, these results indicate that the Venus reporter recapitulates the expression of the endogenous genes, and they also suggest that the endogenous allele is not grossly disrupted.

In Figure 2, please add images for other strains (CRYAA and ACTB) to demonstrate the specificity of expression.

The Reviewer’s point is well taken. We did not generate stable knock-in lines for the *CRYAA* and *ACTB2* loci. These loci were only for testing knock-in efficiency at the F0 stage. We have included additional staining of *CRYAA* and *Venus* in F0 *CRYAA-T2A-Venus* compared to wildtype larvae (new Figure 1—figure supplement 1). We did not perform staining for *ACTB2* F0s because the knock-in for this gene was embryonic lethal (embryos did not survive hatching), which is consistent with what has been observed in Medaka (Gutierrez-Triana, et al., 2018). This could be due to the sensitivity of actin assembly to any perturbation (e.g., additional amino-acids due to cleavage of the P2A peptide). We have added sentences in the main text to reflect this.

Figure 1D shows that wt and ki bands are detected in the F0 animal, which could be a sign of heterozygosity or mosaicism. Can the authors distinguish and potentially estimate the percentage of mosaicism?

The Reviewer brings up another good point. Using the 3-primer PCR we can roughly estimate the level of mosaicism and heterozygosity in the tail (which is the source of genomic DNA for genotyping) by looking at the level of the knock-in specific band vs the wildtype band. We have sentences to address this:

“Genotyping injected animals using this 3-primer PCR strategy enables a rough estimate of the level of heterozygosity and mosaicism in each animal. This estimate is helpful in selecting highly edited F0 founders for generating stable lines, especially in cases where the introduced insertion does not contain a fluorescent reporter and thereby cannot be visually selected (Figure 1D).”

To prove that the targeted promotor faithfully reflects gene expression, a co-staining of the reporter with endogenous gene expression should be done. For the moment, the HCRs shown in Figure4-suppl2 do not have a lot of added value and should be performed in the KI lines.

We agree with the Reviewer that these are important experiments. We have now performed staining to test co-expression of fluorescent reporter transcript (*Venus*) and the target transcripts (e.g., *ELAVL3, CRYAA*, *NPY*, and *HCRT*) via *in situ* hybridization using hybridization chain reaction (HCR) (new Figure 2F, Figure 4C, Figure 4E, and Figure 1— figure supplement 1). We observe that the *Venus* reporter recapitulates the expression pattern of the endogenous target transcript in the knock-in lines *ELAVL3-T2A-Venus, NPYT2A-Venus*, and *HCRT-T2A-Venus* (in heterozygous animals for all lines). The *Venus* reporter also recapitulates the expression pattern of the target transcript in Venus positive F0 animals targeting the CRYAA locus. We also find that *Venus* expression in each knock-in line is similar to the expression of the target gene in the context of wildtype individuals for *ELAVL3* and *CRYAA* (new Figure 2G and Figure 1—figure supplement 1), which we had previously verified for *NPY*, and *HCRT* (Figure 4—figure supplement 2). Together, these results indicate that the Venus reporter recapitulates the expression of the endogenous genes, and they also suggest that the endogenous allele is not grossly disrupted. We also find that Venus expression is similar to the expression of the target gene in the context of wildtype individuals for each target locus (*ELAVL3*, *CRYAA*, *HCRT* and *NPY*). Together, these results indicate that the Venus reporter recapitulates the expression of the endogenous genes.

Regarding "rather limited numbers of animals"This reviewer found it difficult to get a full view of the experimental flow with regard to animal numbers, as information related to efficiency and lethality was spread out in different places. I would recommend making a merged supplemental table displaying the initial number of eggs injected, the number of surviving embryos, the number of positive embryos, etc. for all lines displayed in the paper. This will give a better view of the experimental setup and the efficiency of the method.

Thank you for this helpful feedback. We have now generated a merged table (Supplementary File 4) with the number of injected embryos, surviving embryos, and Venus positive embryos for each replicate in the paper.

The number of animals should be larger (minimally 50 surviving) to strengthen the efficiency claim.

We agree with the Reviewer that increased embryo number would strengthen the manuscript. Given the embryo survival rates and clutch sizes we achieve even with harem breeding, it is challenging to guarantee >50 embryos survive to hatching for a single round of injections. However, to strengthen our efficiency claims, we have performed additional rounds of independent injection replicates for all injection conditions quantified in the paper (with the exception of *ACTB2-T2A-Venus* knock-in line as this knock-in was embryonic lethal). We have now conducted >30 separate rounds of injections and injected >3000 embryos. With the additional replicates, we have between 68–181 total embryos per injection condition that survive two weeks post injection. Because some surviving embryos go into diapause, of the surviving embryos, we have between 43–156 total developed embryos per transgenic line.

We have included all these data in Supplementary File 4 and in modified Figure 1C, Figure 3B, and Figure 1—figure supplement 2.

Germline TM was only shown for 1 line. Did the other lines go germline as well? Does the KI have an effect on fertility?

This is another important point. We have generated stable lines for four different knock-in lines in this paper: *ELAVL3-T2A-Venus*, *ELAVL3*-*T2A*-*Venus*-*P2A*-*H2B*-*oScarlet*, *NPY-T2AVenus*, and *HCRT-T2A-Venus*. We have now indicated this more clearly in the manuscript discussion. We have not carefully tracked fertility in knock-in lines. However, we observe no obvious differences in fertility for these lines, e.g., we do not struggle to maintain these lines. We have now indicated this in material and methods.

Lethality is high; at what stage did the authors lose the embryos?

In our hands, lethality is high in embryos in the first three days post fertilization for both knockin animals and matched wildtype animals. We have included plots of the dynamics of embryo death post fertilization for both injected and matched non-injected animals (from the same collection) (Figure 1—figure supplement 2C). We have also added sentences in the main text regarding this death and recommendation for best practice:

“In general, the lethality of killifish embryos, whether injected or not, was variable in the first two weeks of development (dependent on breeders and clutch) (Figure 1— figure supplement 2B, C). Thus, to ensure successful generation of transgenic animals that survive past development, we recommend injecting ~100 embryos per line.”

Regarding " Figures of the manuscript contain information that is not discussed in the main text"Figure 1D: bands seem to suggest that some F0 are homozygous? What is the percentage of ho in F0? 85% of the fish are homozygous after crossing 2 different F0: these percentages suggest that the F0 might already be edited on both alleles. Editing on both alleles might also have created different additional small edits that, when combined, have a potentially larger effect on gene expression. Have the authors seen such effects in the combined F1 generation?

The Reviewer raises a series of important points. Yes, we believe that some F0 are indeed homozygous at the targeted locus, but we do not know the exact percentage of homozygous in F0. The 3-primer PCR gives a rough estimate of the level of heterozygosity and mosaicism, which can guide selection of founders, but is not an accurate readout of the percentage of homozygous F0s. Our example crossing of 2 F0s resulting in homozygous F1s supports the presence of homozygous F0s, though we have not systematically examined it.

The Reviewer brings up a good point about the risk of combining F0 mutations. We have performed sequencing for stable lines generated in this paper and have not observed mutations in the regions we checked. We agree that crossing F0s together could risk accumulation of mutations if they did exist and thus, we have added further discussion of this in the main text of the paper:

“This accelerated inter-crossing approach could enable rapid testing of homozygous F1 lines if desired. However, we also note that directly generating homozygous F1 lines may also increase the risk of propagating silent mutations from F0 animals (e.g., off-target genome editing in F0s), which might in turn lead to phenotypes independent of the introduced transgene. […] to limit the risk of off-target editing, we recommend outcrossing founders to wildtype animals.”

Figure 2E: green signal apparent in WT in the somites? This is not apparent in all pictures, but I think the authors might have an explanation for this green signal.

Thank you for this suggestion. Unfortunately, we do not know the identity of this green signal. This signal is visible in both the wildtype and the *ELAVL3-T2A-Venus* images in Figure 2E but clearer in wildtype. We believe this difference is due to slight differences in the sagittal section depth of the *ELAVL3-T2A-Venus* animal vs the wildtype. With the small size of the larval killifish, it is difficult to get identically placed sagittal sections. We feel confident that the green signal seen in the transgenic animal is Venus fluorescence given the strength of the signal relative to wildtype. We have now indicated more clearly the green signal in figure legend and indicate that we believe this is autofluorescence:

“In both *ELAVL3-T2A*-*Venus* and wildtype individuals, we observe background green signal for example in the intestine and ventral to the spinal cord. This background signal is not identical between *ELAVL3-T2A*-*Venus* and wildtype samples likely due to slight differences in depth of the sagittal slice.”

Figure3D/E: ELAVL3 is expected to drive expression in all postmitotic neurons. Expression is expected to be similar in all postmitotic neurons, however distinct non-expressing stripes can be observed. This seems to indicate a mosaic situation with some (clonally related?) cells not expressing ELAVL3, but it is unclear how this can appear in an F1 animal (D). Do the authors have an explanation for this phenomenon?Figure 3E Expression in brain hemispheres is not identical is some regions; This is an F0 animal that could have mosaicism; can this be repeated in F1 or F2? Why the different ages? Can the age of the animal have an effect on the expression levels of ELAVL3?

We thank the Reviewer for their very insightful observation. We believe these “non-expressing strips” observed in the Figure 3 are due to mosaicism in F0 animals, with clonal expansion of progenitor cells not expressing *Venus* or *oScarlet*. We have added the following text to the figure legend:

“Strips in the PGZ that appear to lack oScarlet expression could be due to mosaicism of the F0 individual.”

This observation is difficult to understand in the F1 animal shown in Figure 3D. Thus, we double checked our records for this slice and found that in fact the Reviewer was completely correct—this slice is actually from an F0 animal and we had made a mistake in the paper. We have corrected this mistake throughout the paper. We have also double checked all other images to ensure they are annotated correctly. In addition, we have now included additional staining of F3 animals for the same *ELAVL3* locus with the *ELAVL3-T2A-Venus* knock-in line for both *Venus* and *ELAVL3* (new Figure 2F). We do not observe strips of non-expressing cells in these stable lines. So again, we believe this is further evidence suggesting the nonexpressing strips observed on Figure 3 are due to mosaicism of the F0 individual.

In Figure 3E, the three-dimensional rendering makes regions of the brain in the foreground partially obstruct regions in the background to give the appearance of volume. Therefore, some brain hemispheres look dimmer because of the 3D rendering. We included z-slices of select regions to combat this, but feel it is valuable to keep the 3D volume as it still provides information about the distribution of neurons throughout the brain.

We have not done a detailed study of the effect of age on the expression of *ELAVL3*. While this would be very interesting, we believe this is beyond the scope of the knock-in method that we are presenting in this paper. We have indicated in the discussion:

“However we have not examined if the expression level of *ELAVL3* changes with age in this study”.

Regarding: "Full disclosure of methods"The authors mention chemical modifications of the templates #1 and #3 but no more information was given. The IDT website does not mention these modification options, so it will be advisable to reveal the modification or use the appropriate IDT reference number. Used solution recipes should be complete.

Thank you for the feedback. We have now included a step-by-step protocol with all reagents and recipes including the IDT reference number (when available) and product names (new Supplementary File 1). We have also included additional details in the material and methods section. Furthermore, to avoid confusion about reagent information, we no longer refer to the chemical modification on the HDR template as “modification #3”, as this is not the term used for the commercially available product. We now refer to it throughout the revised manuscript as “Alt-R HDR Donor Block” from IDT (and we provide the URL for ordering this type of HDR template from IDT). We now also provide the catalog number for the small molecule HDR enhancer from IDT.

Reviewer #3 (Recommendations for the authors):I recommend this manuscript for publication but have a few questions/comments after reading the paper.1) I think the paper would benefit from putting the work in context with the GeneWeld paper from Wierson et al. 2020 and the method described by Seleit et al. 2022. Contrasting the advantages and disadvantages of this work with theirs would be helpful.

Thank you for this helpful suggestion. We agree that this is a valuable addition to the paper. We have expanded the discussion to include details of advantages and disadvantages of our presented method in the context of these key papers.

2) Can the authors comment on why they used 150-200 bp homology arms?

This is another great suggestion. We selected homology arm length based on IDT recommendations (https://www.idtdna.com/pages/products/crispr-genome-editing/alt-r-hdrdonor-blocks). IDT recommend a range of lengths 100–300 bp for Alt-R HDR Donor Blocks. We selected a length on the shorter end of that scale such that the dsDNA HDR template was shorter and thus less expensive for synthesis. We have now included this information in both the main text and the material and methods.

3) I am left wondering how much of the efficiency reported in this paper is due to the locus being targeted. The presence of a PAM sequence near the endogenous stop codon is a limitation of the technique. Are their ways of circumventing this or alternative strategies? This would be a good discussion point.

The Reviewer brings up an important limitation of CRISPR/Cas9-mediated genome editing approaches. We now more explicitly mention the constraint to the main text:

“The target site for genomic insertion was designed within 2–7 bps of the Cas9 cut sites (Supplementary File 3).”

We added a discussion of this in the methods section of the paper:

“Guide sequences were only selected if followed by the PAM site (5’-NGG-3’) for *Streptococcus pyogenes* Cas9. The necessity of a PAM site near the target insertion site is a constraint of a CRISPR/Cas9-mediated knock-in approach. The Cas9 cut sites were between 2–7 bp from the target insertion site (Supplementary File 3).”

To provide more detail on the flexibility of the knock-in design that is possible within this constraint, we have included the distance between the Cas9 cut site and insertion site for all designs in this paper in Supplementary File 3. We have successfully used gRNAs with cut sites up to 7 bp away from the insertion site (for the *NPY-T2A-Venus* design).

4) On a related note, are the distances between the Cas9 cut sites and target insertion sites reported in the paper?

We agree this would be helpful. We have included the distance between Cas9 cut site and the target insertion site in Supplementary File 3 in the ‘Guides’ tab.

5) It is difficult to interpret supplemental figure 1A since two chemical modifications are used. It would be nice if the panel included modification #3 without HDR enhancer.

The Reviewer is correct that we have not tested modification #3 (now termed “Alt-R HDR Donor Block (IDT)” in the revised manuscript) without the small molecule HDR enhancer. We did not prioritize testing the impact of adding/removing the HDR enhancer because we found that addition of HDR enhancer + modification #3 was better (with statistical significance with additional replicates) than without both, and we continued with the latter condition (revised Figure 1C). That being said, the knock-in efficiency without modification #3 and without the small molecule HDR enhancer is still reasonably good. We have now indicated this more specifically in the Materials and methods and figure legend for Figure 1—figure supplement 2.

6) 1XPBS should have a space. 1X PBS?

Thank you! We have made the change throughout the paper.

References

Bedbrook, C.N., Nath, R.D., Barajas, R., and Brunet, A. (2023). Life Span Assessment in the African Turquoise Killifish *Nothobranchius furzeri*. Cold Spring Harb Protoc.

Freeman, J., Vladimirov, N., Kawashima, T., Mu, Y., Sofroniew, N.J., Bennett, D.V., Rosen, J., Yang, C.T., Looger, L.L., and Ahrens, M.B. (2014). Mapping brain activity at scale with cluster computing. Nat Methods *11*, 941-950.

Gutierrez-Triana, J.A., Tavhelidse, T., Thumberger, T., Thomas, I., Wittbrodt, B., Kellner, T., Anlas, K., Tsingos, E., and Wittbrodt, J. (2018). Efficient single-copy HDR by 5' modified long dsDNA donors. *eLife 7*.

Harel, I., Benayoun, B.A., Machado, B., Singh, P.P., Hu, C.K., Pech, M.F., Valenzano, D.R., Zhang, E., Sharp, S.C., Artandi, S.E., et al. (2015). A platform for rapid exploration of aging and diseases in a naturally short-lived vertebrate. Cell *160*, 1013-1026.

Harel, I., Valenzano, D.R., and Brunet, A. (2016). Efficient genome engineering approaches for the short-lived African turquoise killifish. Nature protocols *11*, 2010-2028.

Hsu, P.D., Scott, D.A., Weinstein, J.A., Ran, F.A., Konermann, S., Agarwala, V., Li, Y., Fine, E.J., Wu, X., Shalem, O., et al. (2013). DNA targeting specificity of RNA-guided Cas9 nucleases. Nat Biotechnol *31*, 827-832.

Kanda, T., Sullivan, K.F., and Wahl, G.M. (1998). Histone-GFP fusion protein enables sensitive analysis of chromosome dynamics in living mammalian cells. Curr Biol *8*, 377-385.

Labun, K., Montague, T.G., Krause, M., Torres Cleuren, Y.N., Tjeldnes, H., and Valen, E. (2019). CHOPCHOP v3: expanding the CRISPR web toolbox beyond genome editing. Nucleic Acids Res *47*, W171-W174.

Nath, R.D., Bedbrook, C.N., Nagvekar, R., and Brunet, A. (2023). Husbandry of the African Turquoise Killifish *Nothobranchius furzeri*. Cold Spring Harb Protoc.

Soriano, P. (1999). Generalized lacZ expression with the ROSA26 Cre reporter strain. Nat Genet *21*, 70-71.